# Observations of organic and inorganic chlorinated compounds and their contribution to chlorine radical concentrations in an urban environment in Northern Europe during the wintertime

Michael Priestley[1], Michael le Breton[1†], Thomas J. Bannan[1], Stephen D. Worrall[1^], Asan Bacak[1], Andrew R. D. Smedley[1*], Ernesto Reyes-Villegas[1], Archit Mehra[1], James Allan[1,2], Ann R. Webb[1], Dudley E. Shallcross[3], Hugh Coe[1], Carl J. Percival[1‖].

[1]Centre for Atmospheric Science, School of Earth and Environmental Sciences, University of Manchester, Manchester, M13 9PL, UK
[2]National Centre for Atmospheric Science, University of Manchester, Manchester, M13 9PL, UK
[3]School of Chemistry, The University of Bristol, Cantock's Close BS8 1TS, UK
[†]Now at Department of Chemistry and Molecular Biology, University of Gothenburg, 412 96 Göteborg, Sweden
^ Now at School of Materials, University of Manchester, Manchester, M13 9PL, UK
* Now at School of Mathematics, University of Manchester, Manchester, M13 9PL, UK
‖ Now at Jet Propulsion Laboratory, 4800 Oak Grove Drive, Pasadena, CA 91109
*Correspondence to:* Carl Percival (carl.j.percival@jpl.nasa.gov)

**Abstract.** A number of inorganic (nitryl chloride, $ClNO_2$; chlorine, $Cl_2$; and hypochlorous acid, $HOCl$) and chlorinated, oxygenated volatile organic compounds (ClOVOCs) have been measured in Manchester, UK during October and November 2014 using time of flight chemical ionisation mass spectrometry (ToF-CIMS) with the $I^-$ reagent ion. ClOVOCs appear to be mostly photochemical in origin although direct emission from vehicles is also suggested. Peak concentrations of $ClNO_2$, $Cl_2$ and $HOCl$ reach 506, 16 and 9 ppt respectively. The concentrations of $ClNO_2$ are comparable to measurements made in London, but measurements of ClOVOCs, $Cl_2$ and $HOCl$ by this method are the first reported in the UK. Maximum $HOCl$ and $Cl_2$ concentrations are found during the day and $ClNO_2$ concentrations remain elevated into the afternoon if photolysis rates are low. $Cl_2$ exhibits a strong dependency on shortwave radiation further adding to the growing body of evidence that it is a product of secondary chemistry, however night time emission is also observed. The contribution of $ClNO_2$, $Cl_2$ and ClOVOCs to the chlorine radical budget suggests that $Cl_2$ can be a greater source of Cl than $ClNO_2$, contributing 74% of the Cl radicals produced on a high radiant flux day. In contrast, on a low radiant flux day, this drops to 14% as both $Cl_2$ production and loss pathways are inhibited by reduced photolysis rates. This results in $ClNO_2$ making up the dominant fraction (83%) on low radiant flux days as its concentrations are still high. As most ClOVOCs appear to be formed photochemically, they exhibit a similar dependence on photolysis, contributing 3% of the Cl radical budget observed here.

## 1. Introduction

Oxidation controls the fate of many atmospheric trace gases. For example, increasing the oxidation state of a given species may increase its deposition velocity (Nguyen et al., 2015) or solubility (Carlton et al. 2006) and

reduce its volatility (Carlton et al. 2006), all of which act to reduce the atmospheric lifetime of that species and can lead to the formation of secondary material such as secondary organic aerosol (SOA) or ozone ($O_3$). As the identity of the chemical species change with oxidation, intrinsic and diverse properties of the chemical species are altered; influencing their toxicity (Borduas et al., 2015) and their impact on the environment e.g. cloud particle nucleating efficiency (Ma et al., 2013) or global warming potential (Boucher et al., 2009).

The hydroxyl radical (OH) is considered the most important daytime atmospheric oxidant due to its ubiquity and high reactivity with an average tropospheric concentration of $10^6$ molecules $cm^{-3}$ (Heal et al., 1995). However, rate coefficients for the reaction of the chlorine radical (Cl) can be two orders of magnitude larger than those for OH (Spicer et al. 1998) indicating that lower Cl concentrations of $1 \times 10^4$ atoms $cm^{-3}$ that are estimated to exist in urban areas (e.g. Bannan et al., 2015), can be just as significant in their contribution to oxidation.

Cl initiated oxidation of volatile organic compounds (VOCs) forms chlorinated analogues of the OH initiated oxidation products, via addition (1) or hydrogen abstraction (2) forming HCl that may react with OH to regenerate Cl. Subsequent peroxy-radicals formed through Cl oxidation can take part in the $HO_x$ cycle and contribute to the enhanced formation of $O_3$ and SOA (Wang and Hildebrandt Ruiz, 2017).

$$R + X \xrightarrow{O_2} R(X)OO^. \tag{1}$$

$$RH + X \xrightarrow{O_2} ROO^. + HX \tag{2}$$

Where X is OH or Cl.

Nitryl chloride ($ClNO_2$) is a major reservoir of Cl that is produced by aqueous reactions between particulate chloride ($Cl^-$) and nitrogen pentoxide ($N_2O_5$) (4, 5). Gaseous $ClNO_2$ is produced throughout the night and is typically photolysed at dawn before $^.OH$ concentrations reach their peak (6). This early morning release of Cl induces oxidation earlier in the day and has been shown to increase maximum 8 hour mean $O_3$ concentrations by up to 7 ppb under moderately elevated $NO_x$ levels (Sarwar et al., 2014). Typical $ClNO_2$ concentrations measured in urban regions range from 10s of ppt to 1000s of ppt. Mielke et al. (2013) measured a maximum of 3.6 ppb (0.04Hz) during summer time in L.A. with maximum sunrise concentrations of 800 ppt. Bannan et al. (2015) measured a maximum concentration of 724 ppt (1Hz) at an urban background site in London during summer. They state that in some instances, $ClNO_2$ concentrations increase after sunrise and attribute this to the influx of air masses with higher $ClNO_2$ concentrations by either advection or from the collapse of the residual mixing layer. In urban environments where $NO_x$ emission and subsequent $N_2O_5$ production is likely, $Cl^-$ may be the limiting reagent in the formation of $ClNO_2$ if excess NO does not reduce $NO_3$ (3) before $N_2O_5$ is produced (e.g. Bannan et al., 2015). Whilst distance from a marine source of $Cl^-$ may explain low, inland concentrations (Faxon et al. 2015), long range transport of marine air can elevate inland $ClNO_2$ concentrations (Phillips et al., 2012) and long range transport of polluted plumes to a marine location can also elevate $ClNO_2$ concentrations (e.g. Bannan et al. 2017).

$$NO_3 + NO \longrightarrow 2NO_2 \tag{3}$$

$$NO_3 + NO_2 \longrightarrow N_2O_{5(g)} \longrightarrow N_2O_{5(aq)} \tag{4}$$

$$Cl^- + N_2O_5 \longrightarrow ClNO_2 + NO_3^- \tag{5}$$

$$ClNO_{2(aq)} \longrightarrow ClNO_{2(g)} \xrightarrow{J_{ClNO_2}} Cl + NO_2 \qquad (6)$$

75 Anthropogenic emission of molecular chlorine is identified as another inland source of $Cl^-$ in the U.S. (e.g. Thornton et al. 2010; Riedel et al. 2012) and in China (e.g. Wang et al. 2017, Liu et al. 2017) where some of the highest concentrations 3.0 - 4.7 ppb have been recorded. As well as industrial processes, the suspension of road salt used to melt ice on roads during the winter has been suggested as a large source of anthropogenic $Cl^-$ (Mielke et al. 2016). This winter time only source, combined with reduced nitrate radical photolysis, is expected 80 to yield greater $ClNO_2$ concentrations at this time of the year (Mielke et al. 2016).

The photolysis of molecular chlorine ($Cl_2$) is another potential source of Cl. Numerous heterogeneous formation mechanisms leading to $Cl_2$ from $Cl^-$ containing particles are known. These include the reaction of $Cl^-$ and OH (Vogt et al., 1996), which may originate from the photolysis of $O_{3(aq)}$ (Oum, 1998) or reactive uptake of $ClNO_2$ (Leu et al., 1995), $ClONO_2$ (Deiber et al., 2004) or HOCl (Eigen and Kustin, 1962) to acidic $Cl^-$ containing 85 particles. Thornton et al. (2010) also suggest that inorganic Cl resevoirs such as HOCl and $ClONO_2$ may also enhance the Cl concentration, potentially accounting for the short fall in the global burden (8-22 Tg $yr^{-1}$ source from $ClNO_2$ and 25-35 Tg $yr^{-1}$ as calculated from methane isotopes). This may be directly through photolysis or indirectly through heterogenous reactions with $Cl^-$ on acidic aerosol.

Globally, $Cl_2$ concentrations are highly variable. In the marine atmosphere, concentrations up to 35 ppt have 90 been recorded (Lawler et al., 2011) whereas at urban costal sites in the US, concentrations on the order of 100s ppt have been measured (Keene et al. 1993, Spicer et al. 1998). Sampling urban outflow, Riedel et al. (2012) measure a maximum of 200 ppt $Cl_2$ from plumes and mean concentrations of 10 ppt on a ship in the LA basin. Maximum mixing ratios of up to 65 ppt have also been observed in the continental US (Mielke et al. 2011).

More interestingly, these studies (Keene et al., 1993; Lawler et al., 2011; Mielke et al., 2011; Spicer et al., 95 1998), report maximum $Cl_2$ concentrations at night and minima during the day. However, there is a growing body of evidence suggesting day time $Cl_2$ may also be observed. Although primary emission may be one source of daytime $Cl_2$ (Mielke et al., 2011), others demonstrate the diurnal characteristics of the $Cl_2$ time series has a broader signal suggestive of continuous processes rather than intermittent signals typically associated with sampling emission sources under turbulent conditions.

In a clean marine environment Liao et al. (2014) observe maximum $Cl_2$ concentrations of 400 ppt attributed to emission from a local snow pack source. A maxima was measured during the morning and evening with a local minimum during mid-day caused by photolysis. They also describe negligible night time concentrations, with significant loss attributed to deposition. Faxon et al. (2015) measured $Cl_2$ with a ToF-CIMS recording a maximum during the afternoon of 4.8 ppt (0.0016 Hz) and suggest a local precursor primary source of $Cl_2$, 105 potentially soil emission, with further heterogeneous chemistry producing $Cl_2$. At a rural site in north China, Liu et al. (2017) measured mean concentrations of $Cl_2$ of 100 ppt and a maximum of 450 ppt, peaking during the day; they also report 480 ppt observed in an urban environment in the US during summer. They attribute power generation facilities burning coal as the source.

Another potential source of Cl to the atmosphere is the photolysis of chlorinated organic compounds (ClVOCs, 110 chlorocarbons, organochlorides) that are emitted from both natural (biomass burning, oceanic and biogenic emission) (e.g. Yokouchi et al. 2000) and anthropogenic sources (e.g. Butler 2000). Whilst many ClVOCs are

only considered chemically important in the stratosphere, those that are photochemically labile in the troposphere e.g. methyl hypochlorite ($CH_3OCl$) whose absorption cross section is non-negligible at wavelengths as long as 460 nm (Crowley et al., 1994), can act as a source of Cl and take part in oxidative chemistry.

Photolysis of ClVOCs have been postulated to contribute 0.1 - 0.5 $\times 10^3$ atoms $cm^{-3}$ globally to the Cl budget of the boundary layer (Hossaini et al., 2016), although on much smaller spatial and temporal scales, the variance in this estimate is likely to be large. Very little data exists on the concentrations, sources and spatial extent of oxygenated ClVOCs (ClOVOCs) and their contribution to the Cl budget.

The ToF-CIMS is a highly selective and sensitive instrument with a high mass accuracy and resolution (m/dm
~4000) that is capable of detecting a suite of chlorinated compounds including HOCl and organic chlorines (Le Breton et al., 2018), as well as other oxygenated chlorine species and chloroamines (Wong et al., 2017). Here we use the ToF-CIMS with the $I^-$ reagent ion to characterise the sources of chlorine and estimate their contribution to Cl concentrations in winter time Manchester, UK.

## 2.    Methodology / experimental

Full experimental details and description of meteorological and air quality measurements can be found in Priestley et al. (2018). A time of flight chemical ionisation mass spectrometer (ToF-CIMS) (Lee et al., 2014b) using iodide reagent ions was used to sample ambient air between 2014-10-29 and 2014-11-11 at the University of Manchester's south campus, approximately 1.5 km south of Manchester City Centre, UK (N53.467, W2.232) and 55 km east of the Irish Sea. Sample loss to the 1m long ¾" PFA inlet was minimised by using a fast inlet
pump inducing a flow rate of 15 standard litres per minute (slm) which was subsampled by the ToF-CIMS. Backgrounds were taken every 6 hours for 20 minutes by overflowing dry $N_2$ and were applied consecutively. The overflowing of dry $N_2$ will have a small effect on the sensitivity of the instrument to those compounds whose detection is water dependent, here we find that due to the low instrumental backgrounds, the absolute error remains small and is an acceptable limitation in order to measure a vast suite of different compounds for
which no best practice backgrounding method has been established. Whilst backgrounds were taken infrequently, they are of a comparable frequency to those used in previous studies where similar species are measured (Lawler et al., 2011; Osthoff et al., 2008; Phillips et al., 2012). The stability of the background responses (i.e. for $Cl_2$ 0.16 ± 0.07 (1σ) ppt) and the stability of the instrument diagnostics with respect to the measured species suggest that they effectively capture the true instrumental background.

Formic acid was calibrated throughout the campaign and post campaign. Very little deviation in the formic acid calibrations was observed. The mean average sensitivity was 30.66 ± 1.90 (1σ) Hz/ppt. A number of chlorinated species were calibrated post campaign using a variety of different methods and relative calibration factors were applied based on measured instrument sensitivity to formic acid as has been performed previously (e.g. le Breton et al. 2014; le Breton et al. 2017; Bannan et al. 2015). A summary of calibration procedures and species
calibrated are described below. All data from between 16:30 on the 5[th] of November to midnight on the 7[th] of November has been removed to prevent the interference of a large scale anthropogenic biomass burning event (Guy Fawkes Night) on these analyses.

### 2.1. Calibrations

We calibrate a number of species by overflowing the inlet with various known concentrations of gas mixtures (Le Breton et al., 2012), including molecular chlorine ($Cl_2$, 99.5% purity, Aldrich), formic acid (98/100%, Fisher) and acetic acid (glacial, Fisher) by making known mixtures (in $N_2$) and flowing 0-20 standard cubic centimetres per minute (sccm) into a 3 slm $N_2$ dilution flow that is subsampled. The $Cl_2$ calibration factor is 4.6 Hz ppt$^{-1}$.

As all chlorinated VOCs we observe are oxygenated we assume the same sensitivity found for 3-chloropropionic acid (10.32 Hz ppt$^{-1}$) for the rest of the organic chlorine species detected. Chloropropionic acid (Aldrich) was calibrated following the methodology of Lee et al. (2014). A known quantity of chloropropionic acid was dissolved in methanol (Aldrich) and a known volume doped onto a filter. The filter was slowly heated to 200$^o$C to ensure total desorption of the calibrant whilst 3 slm $N_2$ flowed over it. This was repeated several times. A blank filter was first used to determine the background.

$ClNO_2$ was calibrated by the method described by Kercher et al. (2009) with $N_2O_5$ synthesised following the methodology described by Le Breton et al. (2014) giving a calibration factor of 4.6 Hz ppt$^{-1}$. Excess $O_3$ is generated by flowing 200 sccm $O_2$ (BOC) through an ozone generator (BMT, 802N) and into a 5 litre glass volume containing $NO_2$ (sigma, >99.5%). The outflow from this reaction vessel is cooled in a cold trap held at -78$^o$C (195 K) by a dry ice/glycerol mixture where $N_2O_5$ is condensed and frozen. The trap is allowed to reach room temperature and the flow is reversed where it is then condensed in a second trap held at 195 K. This process is repeated several times to purify the mixture. The system is first purged by flowing $O_3$ for ten minutes before use. To ascertain the $N_2O_5$ concentration on the line, the flow is diverted through heated line to decompose the $N_2O_5$ and into to a Thermo Scientific 42i $NO_x$ analyser where it is detected as $NO_2$. It is known that the Thermo Scientific 42i $NO_x$ analyser suffers from interferences from $NO_y$ species, indicating that this method could cause an underestimation of the $ClNO_2$ concentrations reported here. Based on previous studies (e.g. Le Breton et al. 2014; Bannan et al. 2017) where comparisons with a broad beam cavity enhancement absorption spectrometer (BBCEAS) have been made, good agreement has been found between co-located $N_2O_5$ measurements. We feel that this calibration method works well, likely in part due to the high purity of $N_2O_5$ synthesised and the possible interference of $NO_y$ on the $NO_x$ analyser during this calibration is considered negligible. The $N_2O_5$ is passed over a salt slurry where excess chloride may react to produce $ClNO_2$. The drop in $N_2O_5$ signal is equated to the rise in $ClNO_2$ as the stoichiometry of the reaction is 1:1. The conversion efficiency of $N_2O_5$ to $ClNO_2$ over wet NaCl is known to vary by between 60-100% (Hoffman et al., 2003; Roberts et al., 2008). Here we follow the methodology of Osthoff et al. (2008) and Kercher et al. (2009) that ensure conversion is 100% efficient and so we assume 100% yield in this study.

We developed a secondary novel method to quantify $ClNO_2$ by cross calibration with a turbulent flow tube chemical ionisation mass spectrometer (TF-CIMS) (Leather et al., 2012). Chlorine atoms were produced by combining a 2.0 slm flow of He with a 0 — 20 sccm flow of 1% $Cl_2$, which was then passed through a microwave discharge produced by a Surfatron (Sairem) cavity operating at 100 W. The Cl atoms were titrated via constant flow of 20 sccm $NO_2$ (99.5% purity $NO_2$ cylinder, Aldrich) from a diluted (in $N_2$) gas mix, to which the TF-CIMS has been calibrated. This flow is carried in 52 slm $N_2$ that is purified by flowing through two heated molecular sieve traps. This flow is subsampled by the ToF-CIMS where the $I.ClNO_2^-$ adduct is measured.

The TF-CIMS is able to quantify the concentration of $ClNO_2$ generated in the flow tube as the equivalent drop in $NO_2^-$ signal. This indirect measurement of $ClNO_2$ is similar in its methodology to $ClNO_2$ calibration by quantifying the loss of $N_2O_5$ reacted with $Cl^-$ (e.g. Kercher et al. 2009). We do not detect an increase in $I.Cl_2$ signal from this calibration and so rule out the formation of $Cl_2$ from inorganic species in our inlet due to unknown chemistry occurring in the IMR. The TF-CIMS method gives a calibration factor 58% greater than that of the $N_2O_5$ synthesis method. The Cl atom titration method and assumes a 100% conversion to $ClNO_2$ and does not take into account any Cl atom loss, which will lead to a reduced $ClNO_2$ concentration and thus greater calibration factor. Also, the method assumes a 100% sampling efficiency between the TF-CIMS and ToF-CIMS; again this could possibly lead to an increased calibration factor. Whilst the new method of calibration is promising, we assume that the proven method developed by Kercher et al. (2009) is the correct calibration factor and assign an error of 50% to that calibration factor. We feel that the difference between the two methods is taken into account by our measurement uncertainty.

We calibrate HOCl using the methodology described by Foster et al. (1999) giving a calibration factor of 9.22 Hz ppt$^{-1}$. 100 sccm $N_2$ is flowed through a fritted bubbler filled with NaOCl solution (min 8% chlorine, Fisher) that meets a dry 1.5 slm $N_2$ flow, with the remaining flow made up of humidified ambient air, generating the HOCl and $Cl_2$ signal measured on the ToF-CIMS. The flow from the bubbler is diverted through a condensed HCl (sigma) scrubber (condensed HCl on the wall of 20cm PFA tubing) where HOCl is titrated to form $Cl_2$. The increase in $Cl_2$ concentration when the flow is sent through the scrubber is equal to the loss of HOCl signal and as the calibration factor for $Cl_2$ is known, the relative calibration factor for HOCl to $Cl_2$ is found.

Additionally, several atmospherically relevant ClVOCs were sampled in the laboratory to assess their detectability by the ToF-CIMS with $I^-$. The instrument was able to detect dichloromethane (DCM, VWR), chloroform (CHCl$_3$, 99.8%, Aldrich) and methyl chloride (CH$_3$Cl, synthesised) although the instrument response was poor. The response to 3-chloropropionic acid was orders of magnitude greater than for the ClVOCs suggesting the role of the chlorine atom is negligible compared with the carboxylic acid group in determining the $I^-$ sensitivity in this case.

## 2.2.    Cl radical budget calculations

Within this system, we designate $ClNO_2$, HOCl and organic chlorine as sources of Cl. As HCl was not detected, it is not possible to quantify the contribution of Cl from the reaction of HCl + OH. Loss processes of Cl are Cl + $O_3$ and Cl + $CH_4$ (7). Photolysis rates for the Cl sources are taken from the NCAR Tropospheric Ultraviolet and Visible TUV radiation model (Mandronich, 1987) assuming 100% quantum yield at our latitude and longitude with column overhead $O_3$ measured by Brewer spectrophotometer #172 (Smedley et al., 2012) and assuming zero optical depth. To account for the effective optical depth of the atmosphere including clouds and other optical components, we scale our idealised photolysis rate coefficient ($J$) by the observed transmittance values in the UV-A waveband (325 to 400 nm). These transmittance values are calculated from UV spectral scans of global irradiance, measured at half-hourly intervals by Brewer spectrophotometer and provided as an output of the shicRIVM analysis routine (Slaper et al., 1995). The Cl rate coefficient for the reaction with $O_3$ is $k_{Cl+O3} = 1.20 \times 10^{-11}$ cm$^3$ molecule$^{-1}$ s$^{-1}$ (Atkinson et al., 2007) and $CH_4$ is $k_{Cl+CH4} = 1.03 \times 10^{-13}$ cm$^3$ molecule$^{-1}$ s$^{-1}$ (Atkinson et al., 2006). The individual $k_{Cl+VOC}$ are taken from the NIST chemical kinetics database.

$$[Cl]_{SS} = \frac{2J_{Cl_2}[Cl_2] + J_{ClNO_2}[ClNO_2] + J_{HOCl}[HOCl] + J_{ClOVOC}\Sigma[ClVOCs]}{k_{O_3+Cl}[O_3] + k_{CH_4+Cl}[CH_4] + \sum_i^n k_{Cl+voc_i}[VOC]_i}$$ (7)

As methane was not measured, an average concentration was taken from ECMWF Copernicus atmosphere monitoring service (CAMS). VOC concentrations were approximated by applying representative VOC:benzene ratios for the UK urban environment (Derwent et al., 2000) and applying those to a typical urban UK benzene:CO ratio (Derwent et al., 1995) where CO was measured at the Whitworth observatory. The

VOC:benzene ratios are scaled to the year of this study to best approximate ambient levels (Derwent et al., 2014). The calculated benzene:CO ratio is in good agreement with a Non-Automatic Hydrocarbon Network monitoring site (Manchester Piccadilly) approximately 1.5 km from the measurement location indicating that the approximation made here is reasonably accurate. The ratios assume traffic emissions are the dominant source of the VOCs as is assumed here.

The photosensitivity of the ClOVOCs to wavelengths longer than 280 nm dictates their ability to contribute to the Cl budget in the troposphere. As many of the identified species here do not have known photolysis rates, we approximate the photolysis of methyl hypochlorite $J_{CH3OCl}$ for all ClOVOCs as it is the only available photolysis rate for an oxygenated organic compound containing a chlorine atom provided by the TUV model and no other more suitable photolysis rate could be found elsewhere e.g. the JPL kinetics database. The same quantum yield

and actinic flux assumptions are made.

## 3.   Results

Concentrations of all chlorinated species are higher at the beginning of the measurement campaign when air masses originating from continental Europe were sampled (Reyes-Villegas et al., 2017). Toward the end of the measurement campaign $ClNO_2$ and ClOVOCs concentrations were low which is consistent with the pollution

during this period having a high fraction of primary components (Reyes-Villegas et al., 2017), see Fig 1.

### 3.1.   Inorganic chlorine

We detect a range of inorganic chlorine species and fragments including $I.Cl^-$, $I.ClO^-$, $I.HOCl^-$, $I.Cl_2^-$, $I.ClNO_2^-$ and $I.ClONO_2^-$ however we do not detect $I.ClO_2^-$, $I.Cl_2O^-$, $I.Cl_2O_2^-$, $I.ClNO^-$ or $I.HCl^-$. Laboratory studies have shown that the ToF-CIMS is sensitive to detection of $I.HCl^-$, however under this configuration, the $I.HCl^-$ adduct

was not observed. The statistics of the concentrations reported below do not take into account the limits of detection (LOD) and so for some of the measurements, values may be reported below the LOD.

### 3.1.1.   $ClNO_2$

$ClNO_2$ (m/z 208) was detected every night of the campaign with a LOD (3x standard deviation of the background) of 3.8 ppt. The 1Hz mean night time concentration of $ClNO_2$ was 58 ppt (not accounting for the

LOD) and a maximum of 506 ppt (not accounting for the LOD) was measured as a large spike on the evening of the 30[th] Oct. These concentrations are comparable to other urban U.K. measured values although the maximum concentration reported here is 30% lower than that measured in London (Bannan et al., 2015) but is consistent with high concentrations expected during the winter as discussed in the introduction.

The diurnal profile of $ClNO_2$ increases through the evening to a local morning maximum with rapid loss after

sunrise. Although we observe a rapid build-up after sunset (ca. 16:30) and loss after sunrise (ca. 07:30), the

maximum concentration measured within a given 24 hour period typically peaks at around 22:00 and halves by 03:00 where it is maintained. The reasons for the early onset in peak concentration and loss throughout the night is unclear although on $1^{th}$ Nov, a sharp decrease in $ClNO_2$ is a consequence of a change in wind direction, indicating the source of $ClNO_2$ is directional. A minimum concentration of <LOD is reached by 15:00 indicating concentrations can persists for much of the day. On $7^{th}$ Nov $ClNO_2$ concentrations grow throughout the morning even after photolysis begins until 11:00. Correlated high wind speeds suggest long range transport and downward mixing is a likely cause for this daytime increase.

Typically, elevated concentrations of $ClNO_2$ are measured when the wind direction is easterly and wind speeds are low (2-4 $ms^{-1}$) and also during periods of southerly winds between 3-9 $ms^{-1}$. The potential sources of $Cl^-$ precursor from these directions are industrial sites, including waste water treatment facilities (8.5 km east and 7.0 km south) that may use salt water as part of the chemical disinfection process (Ghernaout and Ghernaout, 2010). Another source of $ClNO_2$ precursor is found from the south west at wind speeds of 9 $ms^{-1}$ indicating a more distant source which is also likely to be industrial/marine. The correlation between $ClNO_2$ and $Cl_2$ is poor at most times apart from the night of the $30^{th}$ where a strong linear relationship is observed. This is consistent with polluted continental air masses advecting a variety of trace gases. Throughout the measurement campaign the relationship between $ClNO_2$ and $Cl_2$ is poor and so it is unlikely they share the same source.

### 3.1.2. HOCl

HOCl concentrations average 2.18 ppt (not accounting for the LOD) and reach a daytime maximum of 9.28 ppt with an LOD of 3.8 ppt. Concentrations peak in the early afternoon similarly to $Cl_2$ but remain elevated for longer, dropping after sunset. The diurnal profile is similar to that for $O_3$ with a maximum during the day and minima during morning and evening rush hours when $NO_x$ is emitted locally. The strong correlation with $O_3$ ($R^2$ = 0.67) is expected as the route to formation of HOCl is the oxidation of Cl with $O_3$ to form ClO and then oxidation by $HO_2$ to form HOCl. Non negligible night time concentrations of a maximum 8.1 ppt are only measured when concentrations of other inorganic Cl containing species are high. The HOCl signal is artificially elevated after the night of the $5^{th}$ due to a persistent interference from a large scale biomass burning event (Guy Fawkes Night, Priestley et al., 2018) that cannot be deconvolved from the dataset due to the small difference in their mass to charge ratios and insufficient instrument resolution. For this reason HOCl data after this date are discounted from the analysis.

### 3.1.3. ClO

We detect the $I.ClO^-$ adduct at $m/z$ 178 which strongly correlates with $I.ClNO_2^-$ and $I.Cl^-$ signals, all of which show night time maxima. This is inconsistent with the ClO photochemical production pathway of Cl + $O_3$ suggesting its maximum concentration should be measured during the day as was observed for HOCl. It is not possible to confirm if the $I.ClO^-$ is a fragment of a larger ClO containing molecule, however, as the fragmentation of multiple larger molecules are detected as a single adduct e.g. the $I.Cl^-$ cluster is a known fragment from $ClNO_2$ and HOCl, it is reasonable to suspect $I.ClO^-$ may be a fragment as well.

### 3.1.4. $Cl_2$

We observe concentrations of $Cl_2$ during the day ranging from 0 – 16.6 ppt with a mean value of 2.3 ppt (not accounting for the LOD) and night time concentrations of 0 – 4.7 ppt with mean concentrations of 0.4 ppt (not

accounting for the LOD), see Fig 1. The LOD is 0.5 ppt and a calibration factor of 4.5 Hz ppt$^{-1}$ was found. These concentrations are of the same order of magnitude as measured at an urban site in the U.S. but up to 2 orders of magnitude smaller than at U.S urban costal sites (Keene et al. 1993, Spicer et al. 1998) and a megacity impacted rural site in north China (Liu et al., 2017). Although the maximum measured value here is an order of magnitude greater than that measured in Houston (Faxon et al., 2015), the photolysis rate of $Cl_2$ here is two orders of magnitude smaller compared with Houston at that time.

The diurnal profile of $Cl_2$ exhibits a maximum at midday and a minimum at night (early morning) consistent with other studies (Faxon et al., 2015; Liao et al., 2014; Liu et al., 2017). The days with the greatest concentration are those where direct shortwave radiation is at its highest. On the 5$^{th}$ of November, the incidence of direct shortwave radiation is unhindered throughout the day and a similarly uniform profile for $Cl_2$ is also observed. On the 1$^{st}$ of November, $Cl_2$ concentrations increase unhindered as direct radiation increases but when cloud cover reduces radiation transmission efficiency, a corresponding drop in $Cl_2$ is also observed (Fig 2). Equally when global radiation is low throughout the day e.g. 7$^{th}$ November, we observe very low concentrations of $Cl_2$.

There is the potential that the $Cl_2$ signal detected is an instrumental artefact generated either by chemistry in the IMR or from displacement reactions or degassing on the inlet walls. We believe none of these to be the case. First, the correlation between the signal used for labile chlorine in the IMR $^{35}Cl$ (*m/z 35*) is high with $ClNO_2$ ($R^2=0.98$) yet is non-existent with $Cl_2$ ($R^2=0.01$) indicating $Cl_2$ concentration is independent of $^{35}Cl$ concentrations. Second, there is no correlation between $HNO_3$ and $Cl_2$ ($R^2=0.07$) which suggests that acid displacement reactions are not occurring on the inlet walls. Third, there is no correlation between temperature and $Cl_2$ ($R^2=0.08$) indicating that localised ambient inlet heating is also not a contributing factor to increased $Cl_2$ concentrations. Fourth, we observe a similar direct radiation dependency for other photochemical species as we observe for $Cl_2$. For example, the temporal behaviour of $C_2H_4O_5$ exhibits a similar diurnal profile and radiation dependency (Fig 2). Also, the production of $O_3$ increases and decreases with direct solar radiation at the same times we observe the enhancements in concentrations of $Cl_2$ and $C_2H_4O_5$ (Fig 2). The changes in $O_3$ production are observed when NO concentrations are near zero indicating $O_3$ production is VOC limited. Finally, other large organic molecules e.g. $C_{10}H_{14}O_4$ do not exhibit this strong coupling with direct solar radiation. This evidence suggests a local photolytic daytime mechanism is responsible for the increase in daytime concentrations as has previously been suggested (e.g. Finley & Saltzman 2006).

Although peak concentrations of $Cl_2$ are observed in the daytime, high levels of $Cl_2$ are also observed during the night. At the beginning of the measurement period, which has previously been characterised using an aerosol mass spectrometer (AMS) as a period of high secondary activity (Reyes-Villegas et al., 2017), there are persistent, non-zero concentrations of $Cl_2$ ($\leq 4$ ppt) after sunset. On the 4$^{th}$ November, after the period of high secondary activity, intermittent elevations in night time $Cl_2$ concentrations, when the wind is northerly, suggest a local emission source, with concentrations reaching a maximum of 4.6 ppt. Two more distinct night time sources, ranging from the south west through to the east of the measurement site indicate a likely origin of industrial areas, some of which contain chemical production and water treatment facilities.

### 3.2. Organic chlorine

We detected seven $C_2$-$C_6$ ClOVOCs of the forms $C_nH_{2n+1}O_1Cl$, $C_nH_{2n+1}O_2Cl$, $C_nH_{2n+1}O_3Cl$, $C_nH_{2n-1}O_2Cl$, $C_nH_{2n-1}O_3Cl$, $C_nH_{2n-3}O_2Cl$ (Fig 3) of which only $C_2H_3O_2Cl$ has been reported before (Le Breton et al., 2018). We find no evidence for the detection of small chlorohydrocarbons e.g. poly-chloromethanes, such as methyl chloride, dimethyl chloride and chloroform, or poly-chloroethanes such as those described by Huang et al. (2014) in the ambient data, but qualitative testing and laboratory calibrations show that the iodide reagent ion can detect $CH_3Cl$ (not calibrated), $CH_2Cl_2$ (LOD = 143 ppb) and $CHCl_3$ (LOD = 11 ppb). We find no discernible evidence for the detection of 4-chlorocrotonaldehyde, the Cl oxidation product of 1,3-butadiene and unique marker of chlorine chemistry (Wang and Finlayson-Pitts, 2001) due to interferences from other CHO compounds. We do not believe these species are products of inlet reactions as there is a poor correlation ($R^2$ = -0.039) with labile chlorine $^{35}Cl$.

The maximum hourly averaged total ClOVOCs concentration is 28 ppt at 12:00 and at a minimum of 5 ppt at 07:00 when $NO_x$ concentrations are highest at ~30 ppb. Concentrations of $C_2H_3O_2Cl$ (tentatively identified as chloroacetic acid) and $C_6H_{13}OCl$ (tentatively identified as chloro-hexanol) are the highest of any ClOVOCs, accounting for between 20% and 30% respectively of total ClOVOCs concentrations measured. All concentrations rise towards midday with $C_3H_7O_2Cl$ and $C_2H_3O_2Cl$ rising the most by a factor of 4 and returning to nominal levels by the early evening (red in Fig 3). $C_3H_7O_2Cl$ and $C_2H_3O_2Cl$ correlate well with $Cl_2$ ($R^2$ 0.77, 0.75 respectively) which is consistent with a photochemical formation mechanism identifying these species as secondary products, potentially, chloro-propanediol and chloro-acetic acid.

Whilst the diurnal profiles of $C_6H_{13}OCl$ and $C_5H_7O_2Cl$ (blue in Fig 3) are similar to those of $C_3H_7O_2Cl$ and $C_2H_3O_2Cl$, they do not enhance as much as those photochemical species or return to nominal levels after the solar maximum, instead they increase again during the night, with $C_3H_5O_3Cl$ reaching a maximum concentration of 8 ppt at 20:00. This trend suggests concentration changes could be a function of boundary layer height.

$C_3H_7O_2Cl$ and $C_4H_7O_2Cl$ (yellow in Fig 3) are the only ClOVOCs that show a positive correlation with $NO_x$ ($R^2$=0.42, $R^2$=0.41) and negative correlation with $O_3$ ($R^2$=-0.58, $R^2$=-0.53). Their correlation is stronger with $NO_2$ ($R^2$=0.55, $R^2$=0.48), a product of traffic emission. This suggests that at least some of the time, they accumulate at low wind speeds, indicating their origins as local, primary emissions, or as thermal degradation products that have a traffic source e.g. polychlorinated dibenzo-*p*-dioxins/dibenzofurans (PCDD/F) and their oxidation products (Fuentes et al., 2007; Heeb et al., 2013). The diurnal profile shows maxima during mid-day consistent with other photochemical species which is expected of secondary formation. It is possible that these compounds are isobaric or isomeric with other compounds that interfere with the perceived signals recorded here.

The diurnal profile of $C_3H_5O_3Cl$ (green in Fig 3) exhibits a similar shape to the bimodal distribution observed for $NO_x$. Cross-correlation indicates that a time lag of -3 hours provides the best correlation with $NO_2$ of $R^2$=0.80. This is suggestive of local oxidation chemistry taking place over long periods of the day that is sensitive to traffic emission is the source of this ClOVOC.

## 4. Discussion

### 4.1. Effect of global radiation transmission efficiency on Cl radical production

Three days are selected based on their different solar short wave transmission efficiencies to quantify the variation in $Cl_2$ formation and photolysis and so the influence of $Cl_2$ on producing Cl. The average transmission of global radiation on the 5[th] of November was high, $84 \pm 14\%$ (1σ), whereas on the 7[th] of November it was very low $21 \pm 14\%$, sometimes dropping below 10% in the middle of the day. The 1[st] of November serves as a middle case where the transmission efficiency in the morning was high, $88 \pm 11\%$ but in the afternoon was highly variable and dropped to $55 \pm 20\%$ see Fig 4. These three days provide good case studies to investigate the effect of global radiation on molecular chlorine concentrations and therefore the production of Cl.

The reduced transmission efficiency inhibits $Cl_2$ formation thereby reducing the contribution of $Cl_2$ to Cl production. The lower transmission efficiency also reduces the photolysis of $Cl_2$ and so reduces the production of Cl even further. Fig 5. shows the divergence between the ideal $J_{Cl2}$ without transmission efficiency correction (a) and the $J_{Cl2}$ value scaled by transmission efficiency (b) and subsequent Cl formation. Cl production rates are similar until 11am when the scaled production then becomes on average 47% lower. This is most prominent at 13:00 when the difference between ideal and scaled production is $8.4 \times 10^4$ Cl radicals $cm^{-3} s^{-1}$.

### 4.2. Contribution of inorganic chlorine to Cl radical production

The contribution of HOCl and ClOVOCs to Cl formation is negligible due to low photolysis rates and low concentrations whereas the contributions from $Cl_2$ and $ClNO_2$ are much greater (Fig 6). During the morning of the 5[th] Nov, $ClNO_2$ is the dominant source of Cl contributing 95% of total Cl concentration, a maximum of $3.0 \times 10^3$ Cl radicals $cm^{-3}$ to the steady state concentration which is approximately a factor of three lower than the estimated maximum concentration of $9.5 \times 10^3$ Cl radicals $cm^3$ produced by $ClNO_2$ photolysis in London during the summer (Bannan et al. 2015) and a factor of 22 lower than the maximum concentration of $85.0 \times 10^3$ Cl radicals $cm^{-3}$ calculated from measurements of $ClNO_2$ in Houston (Faxon et al., 2015). In both instances this is due to a combination of lower $J_{ClNO2}$ and lower $ClNO_2$ concentrations.

As the day progresses, concentrations of $Cl_2$ increase and it becomes the dominant and more sustained source of Cl contributing 95% of Cl ($12.5 \times 10^3$ Cl radicals $cm^{-3}$) by the early afternoon, which is approximately 4 times that of $ClNO_2$ measured in the early morning and 1.3 times higher than the maximum estimated concentration calculated from $ClNO_2$ photolysis in London (Bannan et al., 2015). The maximum Cl concentration produced from $Cl_2$ and $ClNO_2$ photolysis on the 5[th] reached $14.2 \times 10^3$ Cl radicals $cm^3$ at 11:30 am which is approximately 16% of the $85.0 \times 10^3$ Cl radicals $cm^3$ maximum calculated value from the photolysis of these two species in Houston in summer (Faxon et al., 2015). This is dominated by the contribution of $Cl_2$, indicating $Cl_2$ can be a much more significant source of Cl than $ClNO_2$. On this high flux day, when hourly mean $Cl_2$ concentrations range between $0 - 7$ ppt, the source term is calculated to be between $4 - 21$ ppt $Cl_2$ $hr^{-1}$, which is slightly lower although consistent with previous studies (Faxon et al., 2015; Finley and Saltzman, 2006; Spicer et al., 1998).

The 7[th] Nov has been highlighted as a day with low photolysis rates and high day time $ClNO_2$ concentrations. On this day, $ClNO_2$ is the dominant Cl source (95%) reaching a maximum of $3.4 \times 10^3$ Cl radicals $cm^{-3}$ at 9:30

which is ~87% of that calculated for London (Bannan et al., 2015). A mean $Cl_2$ concentration of 0.3 ppt (less than the LOD of 0.5 ppt) on this day is very low as production of $Cl_2$ at its maximum, calculated as 0.6 ppt hr$^{-1}$, is also low. This combined with a low maximum $J_{Cl2} = 1.13 \times 10^{-4}$ hr$^{-1}$ means maximum Cl production from $Cl_2$ photolysis on this day is very low, generating $0.9 \times 10^3$ Cl radicals cm$^{-3}$ at 10:00 or a quarter of the maximum contributed by $ClNO_2$ on this day, see Fig 6.

$$Cl^-_{(aq)} \xrightarrow{\ J\ } \frac{1}{2} Cl_{2(g)} \xrightarrow{\ J_{Cl2}\ } Cl \tag{8}$$

The dependency of Cl formation on $Cl_2$ production and loss highlights the sensitivity of this reaction channel to photolysis is demonstrated on these two days. The production of Cl from $ClNO_2$ is relatively speaking, less sensitive to the solar flux as the production of $ClNO_2$ does not rely on photochemistry but chemical composition cf. (6) and (8). This further highlights the role of photolytic mechanisms in the re-activation of particulate chloride to gaseous chlorine radicals.

### 4.3.     Organic vs. inorganic contribution to Cl radical production

Summing the concentrations of the ClOVOCs and assuming a uniform photolysis rate $J_{CH3OCl}$ as detailed in the above section, we derive the contribution of total measured ClOVOC to the Cl budget and compare it to the contribution from inorganic Cl measured here (Fig 6). On the high flux day, the Cl concentration reaches $4.0 \times 10^2$ Cl radicals cm$^{-3}$ at midday, which is 30% of the contribution by $ClNO_2$, 3.6% of the contribution from $Cl_2$ and 2% of the HOCl contribution for the same day. On the low flux day, the ClOVOC contribution is $11.0 \times 10^2$ Cl radicals cm$^{-3}$, which is ~2.8% of the $ClNO_2$ contribution on that day and ~57% of the $Cl_2$ contribution. Like $Cl_2$, the production of most ClOVOC requires a photolytic step to generate concentrations that can then go on to decompose providing the Cl. Here it is suggested that the organic contribution to Cl production is negligible at 15% on the low radiant flux day and 3% on the high flux day.

### 5.   Conclusion

A large suite of inorganic and organic, oxygenated, chlorinated compounds has been identified in ambient, urban air during the wintertime in the UK. Of the 7 organic chlorinated compounds (ClOVOCs) identified here, only $C_2H_3O_2ClO$ (tentatively assigned as chloroacetic acid) has previously been reported. Although the ToF-CIMS with I$^-$ is sensitive towards chlorinated and polychlorinated aliphatic compounds e.g. methyl chloride ($CH_3Cl$) dimethyl chloride ($CH_2Cl_2$) and chloroform ($CHCl_3$), their concentrations were below the detection limit. The sources of ClOVOCs are mostly photochemical with maxima of up to 28 ppt observed at midday, although $C_3H_7O_3Cl$ and $C_4H_7O_2Cl$ concentrations correlate with $NO_x$ accumulating at low wind speeds, indicating they are produced locally, potentially as the thermal breakdown products of higher mass chlorinated species such as polychlorinated dibenzo-$p$-dioxins/dibenzofurans (PCDD/F) from car exhausts or the oxidation products thereof. $C_3H_5O_3Cl$ shows a good diurnal cross-correlation with $NO_2$ with a time lag of three hours, suggesting its production is sensitive to $NO_x$ concentrations on that time scale.

Alongside ClOVOCs, daytime concentrations of $Cl_2$ and $ClNO_2$ are measured reaching maxima of 17 ppt and 506 ppt respectively. $ClNO_2$ is a source of Cl throughout every day time period measured. $Cl_2$ shows strong

evidence of a daytime production pathway limited by photolysis as well as emission sources evident during the evening and night time.

On a day of high radiant flux (84±14% of idealised values), $Cl_2$ is the dominant source of Cl, generating a maximum steady state concentration of $12.5 \times 10^3$ Cl radicals $cm^{-3}$ or 74% of the total Cl produced by the photolysis of $Cl_2$, $ClNO_2$, HOCl and ClOVOC with the latter three contributing 19%, 4% and 3% respectively. This contrasts with a share of 14% for $Cl_2$, 83% for $ClNO_2$ and 3% for ClOVOCs on a low radiant flux day (21±14% of idealised values). On the low radiance day not only is the photolysis of all Cl species inhibited,

reducing Cl concentrations, but also the formation of $Cl_2$ and some ClOVOCs by photochemical mechanisms is inhibited thus the variability in contribution between days is highly sensitive to the incidence of sunlight. This further highlights the importance of photochemistry in the re-activation of particulate chloride to gaseous chlorine radicals. Similarly to $Cl_2$, ClOVOCs can be an important source of Cl although the behaviour of their contribution is similar to $Cl_2$ relying on high rates of photolysis, rather than high concentrations as is the case for

$ClNO_2$.

The contribution of the ClOVOCs to the Cl budget would be better determined if more specific photolysis rates for each compound were available and so would further improve the accuracy of the contribution they make to the Cl budget. In addition, future work should aim to identify the processes leading to the formation of these compounds to better constrain the Cl budget in the urban atmosphere. Further ambient measurements of a

broader suite of chlorinated species, as shown here, in different chemical environments would help to better constrain the contribution that chlorine-initiated chemistry has on a global scale.

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

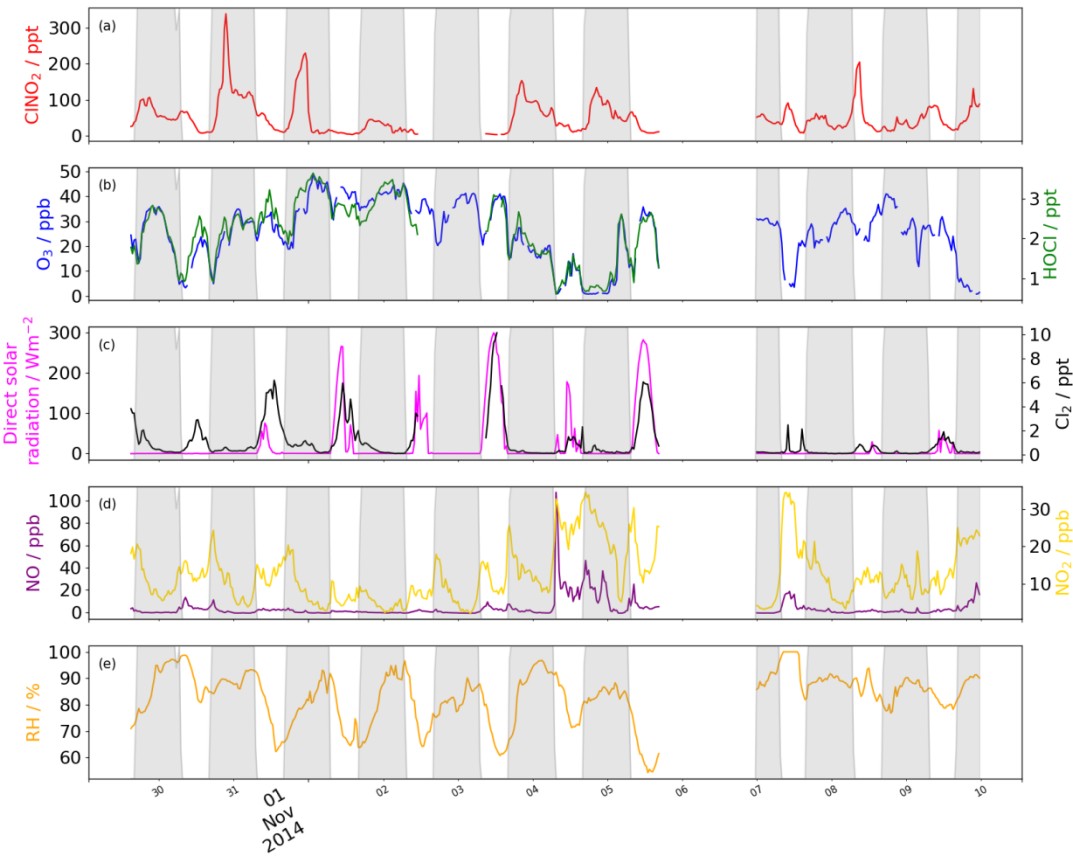

**Fig 1. Time series of a. ClNO$_2$ (ppt). b. HOCl (ppt) and O$_3$ (ppb). c. Cl$_2$ (ppt) and direct solar radiation (Wm$^{-2}$). d. NO (ppb) and NO$_2$ (ppb). e. Relative humidity (%). Data is removed during bonfire night (5[th]-6[th]) and HOCl data is discounted thereafter due to a persistent interference which was not present earlier.**

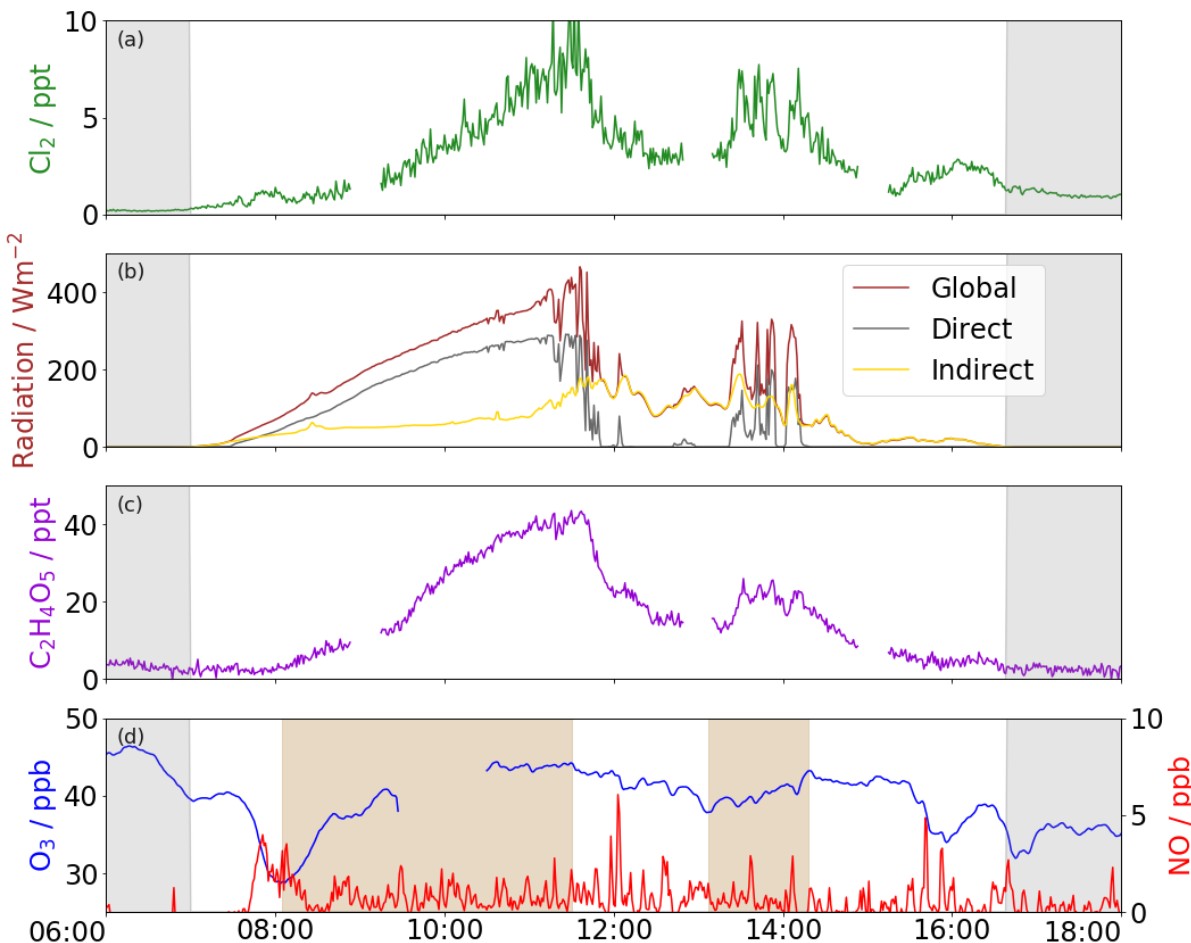

**Fig 2. Time series for the day of 1st Nov 2014. a. Cl$_2$ b. solar radiation (global, direct and indirect) c. photochemical marker C$_2$H$_4$O$_5$ d. O$_3$ and NO$_x$ where highlighted boxes demonstrate $\frac{\Delta[O_3]}{\Delta t}$ is increasing. The increase in concentration of Cl$_2$, C$_2$H$_4$O$_5$ and O$_3$ production when VOC limited are strongly coupled to direct solar radiation. Greyed areas are night time.**

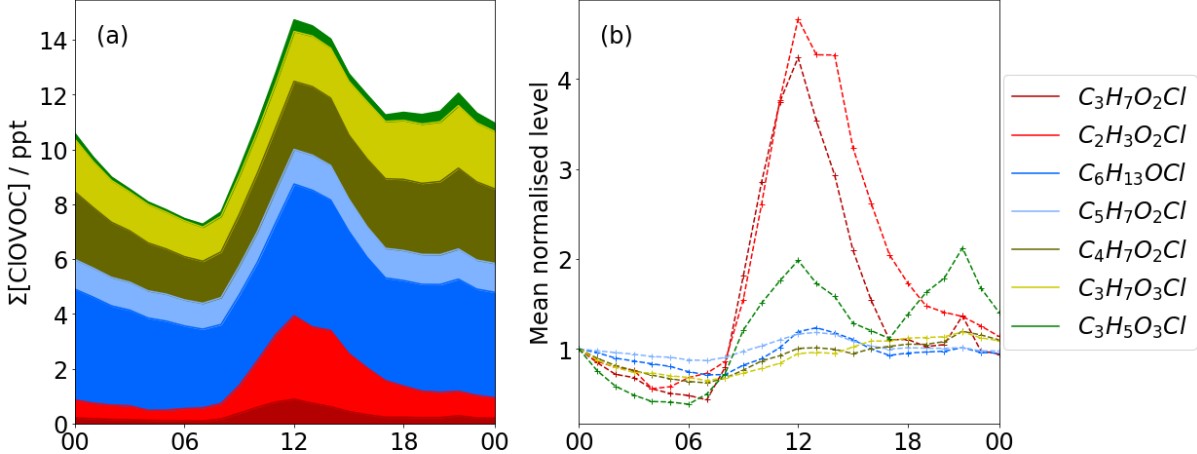

**Fig 3. Diurnal profiles of Cl VOCs. a. Stacked plot showing total Cl VOC concentration. b. The first data point of each diurnal trace is mean normalised to 1.0. Reds show photochemical dominated signals with maxima at midday whereas yellow and blue traces show a more typical diurnal concentration profile associated with changes in boundary layer height indicating these species have longer lifetimes.**

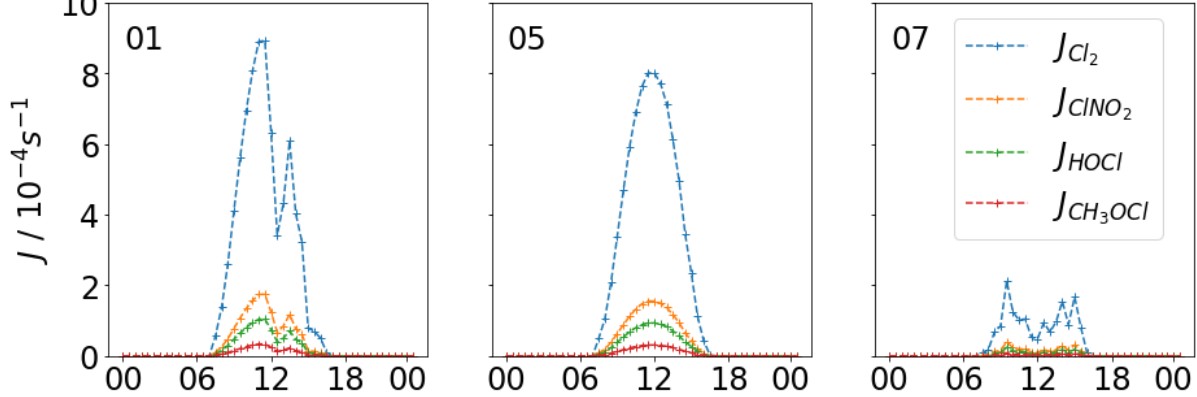

**Fig 4. Transmission scaled J values for $Cl_2$, $ClNO_2$ and HOCl for the 1st, 5th and 7th of Nov. The 1st had high photolysis rates in the morning that were reduced during the afternoon. The 5th is the closest to a full day's ideal photolysis. The 7th shows very weak photolysis.**

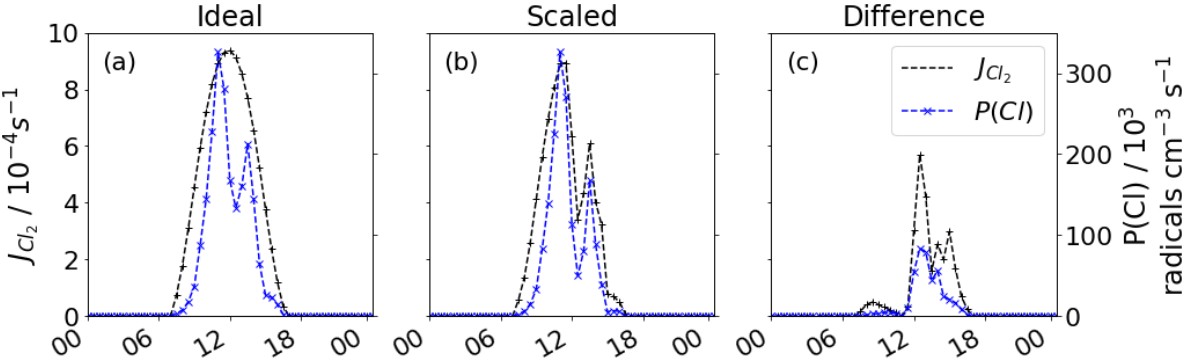

**Fig 5. Diurnal profile for 1$^{st}$ Nov of: a. idealised $J_{Cl2}$ and P(Cl), b. scaled $J_{Cl2}$ and P(Cl), c. the difference between a and b. Transmission efficiency scaled photolysis reduce P(Cl) from $Cl_2$ photolysis.**

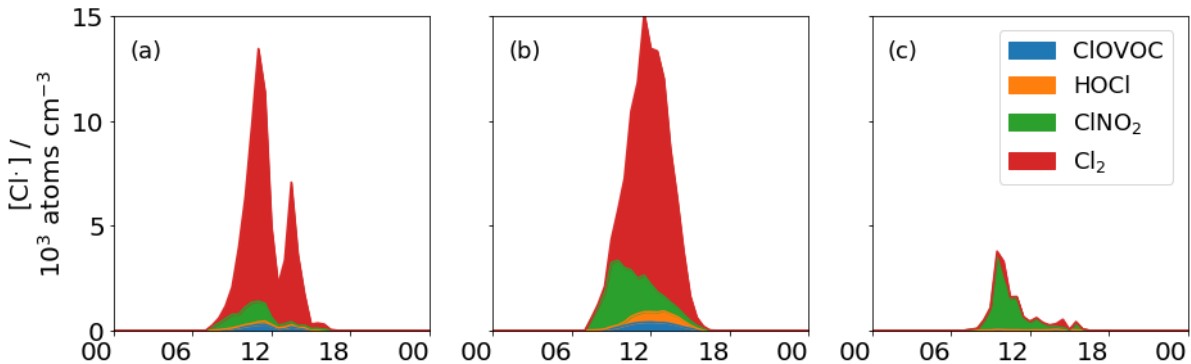

**Fig 6. Steady state concentration of Cl from $ClNO_2$, $Cl_2$, HOCl and total ClOVOC photolysis for a. 1$^{st}$ Nov, b. 5$^{th}$ Nov, and c. 7$^{th}$ Nov. The importance of $ClNO_2$ during the morning is most evident on the 5$^{th}$ with a diminishing contribution throughout the day. On the high flux days, $Cl_2$ is the most important source of Cl but on the low flux day $ClNO_2$ is most important.**