# Peer review of "Observations of organic and inorganic chlorinated compounds and their contribution to chlorine radical concentrations in an urban environment in Northern Europe during the wintertime"

_Atmospheric Chemistry and Physics, 2018_

## Referee Comment (RC1) · Anonymous Referee #1 · 12 Mar 2018

This manuscript summarizes measurements made of organic and inorganic chlorinated species in Manchester, UK. The manuscript focuses on the contribution on ClNO2, Cl2, HOCl and organic chlorides on the daytime Cl radical budget. While the paper is clearly written, there are several shortcomings that need to be addressed prior to publication in ACP. Chiefly among them is an extensive laboratory analysis to validate the identity and calibrate the species presented in this work. As written, most of the paper, with the exception of ClNO2, Cl2, and HOCl observations, could be called speculative and qualitative; however it is presented as quantitative data. The

topics presented in this manuscript could be a relevant and a beneficial contribution to available literature, however the methods are in need of improvement. My recommendation for this manuscript is to return the work to the authors for major revisions including addition work. This manuscript needs laboratory validation of new species, real calibrations, and true error analysis for all species discussed.

The point of this paper is to evaluate the balance of sources of daytime chlorine radicals as observed in Manchester, UK. As such, it becomes important to both state accurate metrics of accuracy and precision in order to truly determine if the radical contributions from a given source are indeed significant. As such, proper zeroing methodology, calibration, and robust error analysis are required in this analysis, all of which seem lacking.

The zeroing method used in this analysis is to zero with dry nitrogen once every 6 hours. The frequency of zeroing is completely insufficient to capture any real changes in instrument background that may have been occurring that are likely to occur on the time scale of minutes to tens of minutes with changes in atmospheric composition, ambient temperature, humidity, etc. Additionally, the iodide adduct ionization scheme is heavily dependent on the water mixing ratio in the flow tube/IMR. In particular, the sensitivity of formic acid and Cl2 are extremely water dependent, in opposite directions, especially at very low IMR water mixing ratios. Considering a typical operating scheme where the inlet flow is matching the source flow, adding dry air at the inlet will result in a 50% change in the IMR humidity. Therefore, dry N2 zeros used here are not truly reflective of the actual ambient background. How were the zeros applied to the data, linear interpolation? What is the error on consecutive zero values? What is the approximate measurement error induced by both the changes in sensitivity due to dry N2 use and interpolation of infrequent zeros. Not accounting for zeros properly will often induce individual features and a diurnal profile that are impossible to distinguish from true ambient observations. Especially when there are humidity dependent changes in sensitivity that then have a diurnal shape related to ambient RH.

One of the major issues with the analysis provided here is a lack of true quantification of many species. The authors state that acetic acid was used to provide a sensitivity for the organic chlorine species. Why would the authors expect that the sensitivity of acetic acid is even relatable to a halogenated organic? The addition of a large electronegative group to the molecule could either reduce the sensitivity due to steric reasons or increase the sensitivity due to increased polarizability or dipole moment. The way I read this manuscript is that one third of the results, particularly the newest results are driven by the observations, quantification, and calculation of the impact of these organic chlorine species on the chlorine radical budget. It seems necessary that for validation and calibration purposes, several of these compounds be bought/made and sampled on the CIMS. The results presented in section 4.3 discussing the potential importance of these species uses a seemingly random calibration factor and an assumption of a uniform photolysis rate from CH3OCl for the calculation. Therefore, data presented for the organic chloride species are completely based on assumptions and not hard quantitative information about the particular molecules being discussed. There is no way of determining the actual error on these numbers, therefore no way to determine if these species have a measurable impact on the Cl radical budget.

The calibration of the N2O5 source, upon which the ClNO2 calibration is determined, seems to need additional laboratory work. The synthesis method, based on Kercher 2009 almost always produces a significant portion of HNO3 which would be detected on both the CIMS and the NOx analyzer. Was there HNO3 observed on the CIMS from the N2O5 source?

The comparison of the two methods of ClNO2 calibration showing a difference of 58% does not tell you anything about your measurement uncertainty, only that one of your calibration methods was flawed or incorrect. Again, without a true robust calibration with known errors it is very difficult to determine the relevance of the numbers calculated in this manuscript. Since this calibration was used to establish the ClNO2 concentration, it is necessary to determine the reasons for the differences in the two

calibration techniques and determine the most accurate calibration method to use, as well as the true instrumental errors.

It is very likely a bad assumption that ClONO2 has the same sensitivity as ClNO2. ClONO2 is likely to have more fragmentation pathways in the IMR than ClNO2.

How do the authors know that the correlation of I2NO2- and NO2 is not indicating that the I2NO2 product is simply some ion formed from various components of NOx. Did the authors add NO2 to the instrument to validate the presence of the I2NO2 cluster ion? If so that should give you a calibration factor, do the concentrations match up? What if the sensitivity is nonlinear, can't that explain the difference and not necessarily a NOy fragmentation product? Why is this section even in this manuscript? What does this have to do with the observations and impact of chlorinated species? In my opinion, all of section 2.3 should be removed from this manuscript and is only a distraction.

The iodide system is based on detection via clustering, so I would assume that this particular instrument would be tuned to promote the formation and stability of ion clusters. As such, the authors need to reconsider the identification of both ICIONO2 and IC2H4O5. If the authors truly are observing ClONO2 that would be one of the first if not the first boundary layer tropospheric observations of that species and would be of significance. I would offer this alternative that ICINO2 is known to fragment to ICl under varying IH2O conditions. That ICl can the cluster with NO3 in the ion flow tube to produce ICINO3. Given the correlation between ClNO2 and ClONO2 in figure 2, there is a significant possibility this is what is occurring. Deviations in the correlation could be driven by changing NO3- levels. I can also contribute that from my experience, heating and cooling of Teflon lines used in source nitrogen will change the amount of Cl- in the ion flow tube. In the case of C2H4O5 the simple explanation without evoking a rare molecule is this is formed during the daytime from the cluster of acetic acid and ozone. This peak is always identified and used in the high res mass list in my own analysis and should be universally included in Iodide adduct mass lists. I encourage the authors to return to the lab and investigate these alternative possibilities prior to publishing this

work.

On the topic of the N2O5 interference, C2H4O5, why is it that the authors indicate that they cannot measure N2O5 during the day, but do not reciprocate the interference. The unknown C2H4O5, which I believe is acetic acid and O3, could be present in the evening as well rendering the N2O5 nighttime measurement suspect. The high-resolution fitting routine should be able to provide some degree of separation between these two species, at the very least in a qualitative sense or to bound the potential magnitude of any interference when the routine is run both including and excluding the C2H4O5 peak. In any case, presentation of this N2O5 discussion is also seemingly unnecessary to this manuscript as the measurements are not discussed or presented anywhere in the manuscript. Therefore I would suggest removal of this discussion.

It does not contribute to the manuscript to include the two discussion about DCM and chloroform for seemingly no other reason than to show first detection. If you do not see them, why is this information relevant? You have calibration factors for these and not the organic chloride compounds of interest. You can use the calibration factors and the detection limit calculated here to at least make a statement of the maximum potential concentration in the atmosphere during your study, if you really need to make a statement about these.

What does the sentence "Methyl chloride and chlorovaleric acid were also detected in the laboratory but not quantified" mean? Did you see them in the ambient data? Were they accidentally detected when sampling lab air? Again, why is this information in the manuscript? Why did you look for these if they are not in the atmosphere.

Because of the potential to have a changing Cl- (m/z 35) in the system and ICl- from ClNO2, I have concerns about the validity of the chlorinate organic observations. It is difficult to interpret these signals as ambient observations especially since the zeros are performed so infrequently and no information can be provided indicating the humidity dependence of the sensitivity of those molecules. It is possible that these features

are driven by instrumental conditions and not reflective of the ambient atmosphere. Do the authors have any data showing instrument zeros occurring during the peak of a high concentration incidence, where the zero doesn't also reduce the instrumental Cl-? Basically, do these species zero? It is difficult to tell from figure 2 and 4, but the peak in organic chloride species and minimum in daytime humidity seem to be coincident on a daily basis. That could be indicative of an ion chemistry sensitivity change. The only way to truly tell if these signals are real is to first establish that you can detect the individual molecules, show a lack of humidity dependence or remove any humidity dependence, and properly account for instrument zeros. Otherwise this data is purely speculative and qualitative, and in the worst-case scenario not real ambient observations.

On page 8, line 276 the authors mention a persistent interference on HOCl, what is that interference? How can the authors conclude that the rest of the data is free of said interference? Why can the two peaks not be resolved via high resolution fitting?

---

## Referee Comment (RC2) · Anonymous Referee #2 · 30 Mar 2018

This manuscripts reports measurements of inorganic chlorine species (ClNO2, Cl2 and HOCl) and chlorinated, oxygenated volatile organic compounds (ClOVOC) taken in Manchester, UK using time of flight chemical ionization mass spectrometry. The authors quantify average concentrations of these species, their diurnal profile as well as their contribution to the total chlorine radical budget.

The manuscript is well written and these measurements and data are of interest to the ACP community. However, I have major concerns regarding the quantification of measured species, as detailed in my comments below, which should be addressed

[Figure]

before publication.

General comments

- The sensitivity of the iodide CIMS is known to depend on relative humidity. This needs to be accounted for in quantification of the analyzed species. This means that the authors will need to redo calibrations at different relative humidity and apply adjusted calibration factors accordingly. This is likely to affect the observed diurnal trends as RH generally varies over the course of the day (with temperature)

- The authors measured backgrounds every 6 hours for 20 minutes and state that they calibrated for formic acid throughout the campaign and during the campaign. I request that the authors provide results from these background measurements and calibrations, e.g. as supplemental information, and interpretation thereof. Some of the questions I would like to see addressed are: how much did background concentrations vary? Which background was subtracted when concentrations varied significantly between two background measurements (i.e. was linear interpolation used)? Overall, how large was the effect of background subtraction on the reported species concentrations? How much did the instrument sensitivity (measured as sensitivity to formic acid) change over time? Were any patterns observed in this change over time?

- Instrument background and inlet effects are expected to vary with relative humidity. This should be addressed in the manuscript. At least authors should take some background measurements in the lab at different relative humidity and report observed differences. (They may also be able to do this in the field considering these measurements were taken from their university campus; albeit in a different season.)

- I have major concerns about the quantification of ClOVOC – see additional specific comments below. If quantification (as a first order estimate) is retained in the revised manuscript, the large uncertainty (from assumptions including using the sensitivity of acetic acid) should be stated again in the results and/or discussion section, not just the methods section

Specific comments:

- Line 129: the authors describe how they minimized losses to the sample line. Did they then characterize the losses? What were they?

- Line 142-144: the authors state that they used the sensitivity of acetic acid for the detected (oxygenated) organic chlorine species (ClOVOC). No justification is provided why this may be appropriate, and I am not convinced that it is. They did calibrate for some non-oxygenated organochlorides. What were the sensitivity of those and how much did it differ from that of acetic acid? I recognize that it might be difficult to obtain standards of the oxygenated organochlorides, but could the authors calibrate for one ClOVOC? Again, how does that sensitivity compare to the sensitivity of the other organochlorides and other oxygenated organics? How does it depend on RH (see comment above)? Overall, with the work that has been done so far, I am unconvinced that it is appropriate to quantify ClOVOC. Also, qualitative trends (which are interesting in the absence of calibration) need to be adjusted for RH effects.

- Lines 159-161: the authors state that they "feel" this calibration method works well, but that is not very convincing, also considering that the results differ by 58% compared to the other calibration method applied. They authors then state that they consider this 58% their measurement uncertainty. A few points regarding this are that: 1) the measurement uncertainty is not mentioned anywhere else in the paper but should be - quantitative results should be stated with a measurement uncertainty 2) other variables are expected to increase total uncertainty, including inlet and background effects and changes in instrument sensitivity. These other factors should be included in the total measurement uncertainty.

- Line 164-166: do these previous studies ensure a 100% conversion efficiency (as currently stated) or do they assume it? If they ensure it, how so? If they assume it, what is the justification?

- Line 175: what justifies the assumption that ClONO2 would have the same sensitivity

as ClNO2?

- Line 188-190: the authors state that HCl was not measured; however, discussion later in the manuscript seems to suggest HCl was not detected, which is different. Please clarify.

- Line 190 (loss processes of Cl): Why do the authors not include the VOC + Cl loss mechanism? (Presumably, this is how the observed ClOVOC are formed.)

- Equation 7 – why does this equation not include photolysis of the organochlorides (ClOVOC)?

- Line 203-206: Have the authors checked whether photolysis of other ClOVOC are available in other models/databases, e.g. the JPL kinetics database https://jpldataeval.jpl.nasa.gov/? How do the photolysis rates compare?

- Line 236-238: please justify not accounting for LOD when calculating statistics on the observed concentrations

- Line 267: the authors find that the behaviors of ClONO2 and ClNO2 are very similar. Is this consistent with what we would expect? Could the observed ions be fragments of each other (i.e. only be one species)?

- Line 330-332: what is the LOD for these species?

- Section 4.2: Could you compare these predicted Cl concentrations with predicted OH concentrations (to be able to assess the importance of Cl chemistry compared to OH chemistry)?

- Line 417: what makes the assignment of the molecule tentative?

- Line 435-436. The authors state that ClOVOC can be a source of Cl. Aren't VOCs also a Cl sink? See comment above.
* * *
[Figure]

2018.

---

## Author Comment (AC1) · 28 Jun 2018

We thank the reviewers for their time evaluating this manuscript. The corrections and additions made as a result of these comments have greatly improved the consistency and focus of this work. The response to each point immediately follows each comment and is coloured red. Updated quotations from the manuscript are in blue. Quotes from the original text are in green.

Responses to Referee 1

[Figure]

This manuscript summarizes measurements made of organic and inorganic chlorinated species in Manchester, UK. The manuscript focuses on the contribution on ClNO2, Cl2, HOCl and organic chlorides on the daytime Cl radical budget. While the paper is clearly written, there are several shortcomings that need to be addressed prior to publication in ACP. Chiefly among them is an extensive laboratory analysis to validate the identity and calibrate the species presented in this work. As written, most of the paper, with the exception of ClNO2, Cl2, and HOCl observations, could be called speculative and qualitative; however it is presented as quantitative data. The topics presented in this manuscript could be a relevant and a beneficial contribution to available literature, however the methods are in need of improvement. My recommendation for this manuscript is to return the work to the authors for major revisions including addition work. This manuscript needs laboratory validation of new species, real calibrations, and true error analysis for all species discussed. The point of this paper is to evaluate the balance of sources of daytime chlorine radicals as observed in Manchester, UK. As such, it becomes important to both state accurate metrics of accuracy and precision in order to truly determine if the radical contributions from a given source are indeed significant. As such, proper zeroing methodology, calibration, and robust error analysis are required in this analysis, all of which seem lacking.

The zeroing method used in this analysis is to zero with dry nitrogen once every 6 hours. The frequency of zeroing is completely insufficient to capture any real changes in instrument background that may have been occurring that are likely to occur on the time scale of minutes to tens of minutes with changes in atmospheric composition, ambient temperature, humidity, etc.

Whilst we agree this backgrounding method is not optimal, it does not affect the quality of the measurements presented here. In line with the current best practice
disseminated after this dataset was collected, we now collect regular fast backgrounds. This method is invaluable for e.g. aircraft measurements, but in a stable laboratory environment changes in environmental variables such as temperature are well controlled and so the instrument background stays fairly constant. The stability of the mass calibration parameters (p1 = 1742.34 $\pm$ 0.02 (1$\sigma$) and p2 = -2125.64 $\pm$ 0.18 (1$\sigma$)) indicate sensitivity changes due to the effect of temperature and impedance on the pd within the instrument do not occur.

Within the iodide CIMS system, recent work shows that long backgrounds remove the species adsorbed on the instrument (IMR) and sample line walls as well as the gas phase species present. So whilst not ideal, there is merit in having a long background.

We have followed previously accepted methods where backgrounds have been applied on the order of hourly time scales for measurements of $ClNO_2$ and $Cl_2$ (e.g. Lawler et al. 2011; Osthoff et al. 2008; Phillips et al. 2012) that would also not capture variations on a minute time scale. So whilst we agree that the backgrounding could be considered not optimal, the measurements presented are not deemed to be affected by the methods employed here.

In recognition of this comment we have now added a discussion on this within the text:

"Whilst backgrounds were taken infrequently, they are of a comparable frequency to those used in previous studies where similar species are measured (Osthoff et al. 2008; Lawler et al. 2011; Phillips et al. 2012). The stability of the background responses (i.e. for $Cl_2$ 0.16 $\pm$ 0.07 (1$\sigma$) ppt) and the stability of the instrument diagnostics with respect to the measured species suggest that they effectively capture the true instrumental background."

Additionally, the iodide adduct ionization scheme is heavily dependent on the water mixing ratio in the flow tube/IMR. In particular, the sensitivity of formic acid and Cl2 are extremely water dependent, in opposite directions, especially at very low IMR water mixing ratios. Considering a typical operating scheme where the inlet flow is matching the source flow, adding dry air at the inlet will result in a 50% change in the IMR humidity. Therefore, dry N2 zeros used here are not truly reflective of the actual ambient background.

Although overflowing dry $N_2$ as a method of backgrounding has been performed for iodide CIMS in the past (e.g. Lee et al. 2014; Crisp et al. 2014), we recognise that this method can alter the sensitivity of the instrument to species whose adduct formation is impacted by the presence of water. However, as the power of the ToF-CIMS comes from measuring hundreds of different molecules, many of which do not have a known, reliable backgrounding method, we feel using a dry $N_2$ background is the best way to approximate a reasonable background for the vast majority of measured species. We feel the methodology here is an acceptable limitation in order to better quantify potentially hundreds of other species that are detected.

Formic acid is commonly measured with iodide CIMS. Its adduct formation with iodide exhibits a strong humidity dependency (one of the strongest known). When this molecule is targeted specifically, a sodium bicarbonate scrubber is used to purge ambient air providing a background at ambient humidity and so, the same sensitivity. This technique is used only to remove organic acids and would not effectively scrub other species.

Laboratory tests show (not published) that when using the bicarbonate scrubber, 75% of the formic acid signal is removed relative to ambient laboratory concentrations. When a dry $N_2$ background is used, 90% of the signal is removed. The difference between the two is caused by a reduction in sensitivity as less water is present when

dry $N_2$ is used. We feel the difference between the two techniques, whilst observable, is an acceptable limitation in order to better quantify potentially hundreds of other species that are detected. It is more than likely that these other measured species do not exhibit anywhere near as much of a humidity dependency (or any at all) or have small ambient concentrations such that a larger relative error in their backgrounds only contributes a small absolute error.

We have performed water sensitivity tests and agree that the $Cl_2$ measurements are dependent on humidity within the IMR, however under this setup we find $Cl_2$ sensitivity is enhanced by the presence of water (as has been reported elsewhere (Lee et al. 2014)). In general, we do not believe comparing the responses from different mass spectrometers is a good validation method as the individual instruments and their tuning vary their responses.

We find an enhanced signal of $Cl_2$ (relative to dry conditions (RH = 0%)) of 1.9 – 2.0 for the interquartile range of our ambient RH measurements (error bars on red point). This extends to 1.66 – 2.05 when including maximum and minimum values (red crosses). This suggests that 75% of the measurements may have suffered a +/- 10% change in instrument sensitivity.

The following has been added to the text:

"The overflowing of dry $N_2$ will have a small effect on the sensitivity of the instrument to those compounds whose detection is water dependent, here we find that due to the low instrumental backgrounds, the absolute error remains small and is an acceptable limitation in order to measure a vast suite of different compounds for which no best practice backgrounding method has been established."

How were the zeros applied to the data, linear interpolation? What is the error on consecutive zero values? What is the approximate measurement error induced by

both the changes in sensitivity due to dry N2 use and interpolation of infrequent zeros. Not accounting for zeros properly will often induce individual features and a diurnal profile that are impossible to distinguish from true ambient observations. Especially when there are humidity dependent changes in sensitivity that then have a diurnal shape related to ambient RH.

The zeros are applied consecutively. The variation in backgrounds for $Cl_2$ is shown in the figure 2. The following has been added to the text

"Backgrounds were taken every 6 hours for 20 minutes by overflowing dry N $N_2$ and were applied consecutively."

As demonstrated in figure 1, the instrument response is enhanced by 100 +/- 10% under our ambient water conditions compared with the dry $N_2$ condition. This suggests the backgrounds may be underestimated by 100% or, 0.18 ppt. The $2\sigma$ of the backgrounds is 0.12 ppt. This results in a propagative error of $\sqrt{0.18^2 + 0.1^2}$ = 0.22, or 1.83% of the maximum measured value.

Throughout the measurement period we find the water counts in the IMR are independent of ambient RH indicating that the variation in the reported species cannot be influenced by a changing ambient RH affecting the instrument response (Fig 3). This may be due to the addition of water to the ionisation mixture or the tuning of the instrument.

Furthermore we find the water cluster signal is stable at times of changing $Cl_2$ and $C_2H_3O_2Cl$ (Fig 4) indicating their responses are not dependent on the water cluster signal.

One of the major issues with the analysis provided here is a lack of true quantification of many species. The authors state that acetic acid was used to provide a

sensitivity for the organic chlorine species. Why would the authors expect that the sensitivity of acetic acid is even relatable to a halogenated organic? The addition of a large electronegative group to the molecule could either reduce the sensitivity due to steric reasons or increase the sensitivity due to increased polarizability or dipole moment. The way I read this manuscript is that one third of the results, particularly the newest results are driven by the observations, quantification, and calculation of the impact of these organic chlorine species on the chlorine radical budget. It seems necessary that for validation and calibration purposes, several of these compounds be bought/made and sampled on the CIMS. The results presented in section 4.3 discussing the potential importance of these species uses a seemingly random calibration factor and an assumption of a uniform photolysis rate from CH3OCl for the calculation. Therefore, data presented for the organic chloride species are completely based on assumptions and not hard quantitative information about the particular molecules being discussed. There is no way of determining the actual error on these numbers, therefore no way to determine if these species have a measurable impact on the Cl radical budget.

The authors agree that the use of acetic acid was misleading and that further calibrations were required in order to accurately report the ClOVOC species. As a result of this we have calibrated for 3-chloropropionic acid and find its sensitivity to be 10.32 Hz ppt$^{-1}$ with a relative formic acid calibration factor of 3.34 Hz ppt$^{-1}$. This calibration method is similar to that described in Mohr et al. (2013). These concentration changes have been incorporated into the manuscript and analysis and the following text has been added.

"As all chlorinated VOCs we observe are oxygenated we assume the same sensitivity found for 3-Chloropropionic acid for the rest of the organic chlorine species detected. Chloropropionic acid (Aldrich) was calibrated following the methodology of Lee et al. (2014). A known quantity of chloropropionic acid was dissolved in methanol (Aldrich) and a known volume doped onto a filter. The filter was slowly heated to 200$^{o}$C to

ensure total desorption of the calibrant whilst 3 slm $N_2$ flowed over it. This was repeated several times. A blank filter was first used to determine the background."

The calibration of the N2O5 source, upon which the ClNO2 calibration is determined, seems to need additional laboratory work. The synthesis method, based on Kercher 2009 almost always produces a significant portion of HNO3 which would be detected on both the CIMS and the NOx analyzer. Was there HNO3 observed on the CIMS from the N2O5 source?

In the text we have already discussed the limitations of this method in detail but we will also respond to the comments raised here. We work very hard to reduce the water in the system through purging cycles by flowing $O_3/O_2$ through the apparatus. Whilst we have seen evidence of $NO_y$ on the Thermo Scientific 42i $NO_x$ analyser during inter-comparisons we have performed with a BBCEAS, there is a small error in reported concentrations. We therefore deem this an appropriate calibration method as many others have (e.g. Kercher et al. 2009). In the Bannan et al. (2015) study $HNO_3$ made up 7% of the reported signal on the Thermo Scientific 42i $NO_x$ analyser which is lower than the reported error of that instrument.

The comparison of the two methods of ClNO2 calibration showing a difference of 58% does not tell you anything about your measurement uncertainty, only that one of your calibration methods was flawed or incorrect. Again, without a true robust calibration with known errors it is very difficult to determine the relevance of the numbers calculated in this manuscript. Since this calibration was used to establish the ClNO2 concentration, it is necessary to determine the reasons for the differences in the two calibration techniques and determine the most accurate calibration method to use, as well as the true instrumental errors.

Beyond the traditional CIMS $ClNO_2$ calibration method, here we present an alternative calibration method for $ClNO_2$ using the turbulent flow tube CIMS. We understand the phrasing in the manuscript is poor and address that here. The following clarification

has been added to the text

"We developed a secondary novel method to quantify $ClNO_2$ by cross calibration with a turbulent flow tube chemical ionisation mass spectrometer (TF-CIMS) (Leather et al. 2012). Chlorine atoms were produced by combining a 2.0 SLM flow of He with a 0 - 20 SCCM flow of 1% $Cl_2$, which was then passed through a microwave discharge produced by a Surfatron (Sairem) cavity operating at 100 W. The Cl atoms were titrated via constant flow of 20 sccm $NO_2^-$ (99.5% purity $NO_2^-$ cylinder, Aldrich) from a diluted (in $N_2$) gas mix, to which the TF-CIMS has been calibrated. This flow is carried in 52 slm $N_2$ that is purified by flowing through two heated molecular sieve traps. This flow is subsampled by the ToF-CIMS where the $I.ClNO_2^-$ adduct is measured. The TF-CIMS is able to quantify the concentration of $ClNO_2$ generated in the flow tube as the equivalent drop in $NO_2^-$ signal. This indirect measurement of $ClNO_2$ is similar in its methodology to $ClNO_2$ calibration by quantifying the loss of $N_2O_5$ reacted with $Cl^-$ (e.g. Kercher et al. 2009). We do not detect an increase in $I.Cl_2$ signal from this calibration and so rule out the formation of $Cl_2$ from inorganic species in our inlet due to unknown chemistry occurring in the IMR. The TF-CIMS method gives a calibration factor 58% greater than that of the $N_2O_5$ synthesis method. The Cl atom titration method assumes a 100% conversion to $ClNO_2$ and does not take into account any Cl atom loss, which will lead to a reduced $ClNO_2$ concentration and thus greater calibration factor. Also, the method assumes a 100% sampling efficiency between the TF-CIMS and ToF-CIMS, again this could possibly lead to an increased calibration factor. Whilst the new method of calibration is promising, we assume that the proven method developed by Kercher et al. (2009) is the correct calibration factor and assign an error of 50% to that calibration factor. We feel that the difference between the two methods is taken into account by our measurement uncertainty."

It is very likely a bad assumption that ClONO2 has the same sensitivity as ClNO2. ClONO2 is likely to have more fragmentation pathways in the IMR than ClNO2. How do the authors know that the correlation of I2NO2- and NO2 is not indicating that the

I2NO2 product is simply some ion formed from various components of NOx. Did the authors add NO2 to the instrument to validate the presence of the I2NO2 cluster ion? If so that should give you a calibration factor, do the concentrations match up? What if the sensitivity is nonlinear, can't that explain the difference and not necessarily a NOy fragmentation product? Why is this section even in this manuscript? What does this have to do with the observations and impact of chlorinated species? In my opinion, all of section 2.3 should be removed from this manuscript and is only a distraction.

We agree that this section does not add value to the discussion of the manuscript so have removed it as the reviewer has suggested.

The iodide system is based on detection via clustering, so I would assume that this particular instrument would be tuned to promote the formation and stability of ion clusters. As such, the authors need to reconsider the identification of both IClONO2 and IC2H4O5. If the authors truly are observing ClONO2 that would be one of the first if not the first boundary layer tropospheric observations of that species and would be of significance. I would offer this alternative that IClNO2 is known to fragment to ICl under varying IH2O conditions. That ICl can the cluster with NO3 in the ion flow tube to produce IClNO3. Given the correlation between ClNO2 and ClONO2 in figure 2, there is a significant possibility this is what is occurring. Deviations in the correlation could be driven by changing NO3- levels. I can also contribute that from my experience, heating and cooling of Teflon lines used in source nitrogen will change the amount of Cl- in the ion flow tube.

We agree that without further work, which is beyond the scope of this manuscript, the identification of the source of $ClONO_2$ cannot be definitively corroborated. For this reason, the discussion of $ClONO_2$ has been removed.

In the case of C2H4O5 the simple explanation without evoking a rare molecule is this is formed during the daytime from the cluster of acetic acid and ozone. This

peak is always identified and used in the high res mass list in my own analysis and should be universally included in Iodide adduct mass lists. I encourage the authors to return to the lab and investigate these alternative possibilities prior to publishing this work.

We use I.$C_2H_4O_5$ as a further example (beyond that of $O_3$) as a marker of photochemistry to provide further evidence that a photochemical mechanism is driving the production of $Cl_2$. It is also possible that more than one isomer of $C_2H_4O_5$ contributes to this signal. It is beyond the scope of this work to definitively identify the structure of associated with this formula. For this reason, we have removed references to the exact identification of this mass however the discussion referencing the formula remains in the text as it demonstrates the point that other photochemical markers of photochemistry behave similarly.

On the topic of the N2O5 interference, C2H4O5, why is it that the authors indicate that they cannot measure N2O5 during the day, but do not reciprocate the interference. The unknown C2H4O5, which I believe is acetic acid and O3, could be present in the evening as well rendering the N2O5 night time measurement suspect. The high resolution fitting routine should be able to provide some degree of separation between these two species, at the very least in a qualitative sense or to bound the potential magnitude of any interference when the routine is run both including and excluding the C2H4O5 peak. In any case, presentation of this N2O5 discussion is also seemingly unnecessary to this manuscript as the measurements are not discussed or presented anywhere in the manuscript. Therefore I would suggest removal of this discussion.

We agree the $N_2O_5$ discussion is not necessary and so it has been removed.

It does not contribute to the manuscript to include the two discussion about DCM and chloroform for seemingly no other reason than to show first detection. If you do not

see them, why is this information relevant? You have calibration factors for these and not the organic chloride compounds of interest. You can use the calibration factors and the detection limit calculated here to at least make a statement of the maximum potential concentration in the atmosphere during your study, if you really need to make a statement about these.

These species were calibrated in the lab but not measured in the ambient data. We mention this in the manuscript because we believe this may be useful to the community and may be important where the concentrations of these species are enhanced above their limits of detection. Additionally, we feel it is important to comment on the sensitivity of iodide towards Cl containing molecules. We do not believe these sentiments detract from the manuscript. We have added the following clarification to the text:

"Additionally, several atmospherically relevant ClVOCs were sampled in the laboratory to assess their detectability by the ToF-CIMS with $I^-$. The instrument was able to detect dichloromethane (DCM, VWR), chloroform ($CHCl_3$, 99.8%, Aldrich) and methyl chloride ($CH_3Cl$, synthesised) although the instrument response was poor. The response to 3-chloropropionic acid was orders of magnitude greater than for the ClVOCs suggesting the role of the chlorine atom is negligible compared with the carboxylic acid group in determining the $I^-$ sensitivity in this case."

What does the sentence "Methyl chloride and chlorovaleric acid were also detected in the laboratory but not quantified" mean? Did you see them in the ambient data? Were they accidentally detected when sampling lab air? Again, why is this information in the manuscript? Why did you look for these if they are not in the atmosphere.

Methyl chloride is an atmospherically relevant species. Prior to the measurement campaign it was synthesised and measured in the laboratory. We have added a

clarification to the text as a response to the previous point.

Because of the potential to have a changing Cl- (m/z 35) in the system and ICl-from ClNO2, I have concerns about the validity of the chlorinate organic observations. It is difficult to interpret these signals as ambient observations especially since the zeros are performed so infrequently and no information can be provided indicating the humidity dependence of the sensitivity of those molecules. It is possible that these features are driven by instrumental conditions and not reflective of the ambient atmosphere. Do the authors have any data showing instrument zeros occurring during the peak of a high concentration incidence, where the zero doesn't also reduce the instrumental Cl-? Basically, do these species zero? It is difficult to tell from figure 2 and 4, but the peak in organic chloride species and minimum in daytime humidity seem to be coincident on a daily basis. That could be indicative of an ion chemistry sensitivity change. The only way to truly tell if these signals are real is to first establish that you can detect the individual molecules, show a lack of humidity dependence or remove any humidity dependence, and properly account for instrument zeros. Otherwise this data is purely speculative and qualitative, and in the worst-case scenario not real ambient observations.

We do not believe the features are driven by the instrumental conditions due to the strong linearity of the calibration response, reagent ion response, background response, instrument diagnostics and independence of RH on water as previously discussed.

In the text we have previously rationalised the signal for $Cl_2$ as real by stating the following:

"There is the potential that the $Cl_2$ signal detected is an instrumental artefact generated either by chemistry in the IMR or from displacement reactions or degassing on the inlet walls. We believe none of these to be the case. First, the correlation between the signal used for labile chlorine in the IMR $^{35}$Cl (m/z 35) is high with ClNO$_2$

($R^2$=0.98) yet is non-existent with $Cl_2$ ($R^2$=0.01) indicating $Cl_2$ concentration is independent of 35Cl concentrations. Second, there is no correlation between $HNO_3$ and $Cl_2$ ($R^2$=0.07) which suggests that acid displacement reactions are not occurring on the inlet walls. Third, there is no correlation between temperature and $Cl_2$ ($R^2$=0.08) indicating that localised ambient inlet heating is also not a contributing factor to increased $Cl_2$ concentrations. Fourth, we observe a similar direct radiation dependency for other photochemical species as we observe for $Cl_2$."

By extending this analysis to $C_2H_3O_2Cl$ we find an $R^2$=-0.039 with $^{35}Cl$ (m/z 35) suggesting that the formation of $C_2H_3O_2Cl$ is not occurring from secondary reactions in the IMR. The following has been added to the text:

"We do not believe these species are products of inlet reactions as there is a poor correlation ($R^2$=-0.039) with labile chlorine $^{35}Cl$."

Responses to Referee 2 This manuscripts reports measurements of inorganic chlorine species (ClNO2, Cl2 and HOCl) and chlorinated, oxygenated volatile organic compounds (ClOVOC) taken in Manchester, UK using time of flight chemical ionization mass spectrometry. The authors quantify average concentrations of these species, their diurnal profile as well as their contribution to the total chlorine radical budget. The manuscript is well written and these measurements and data are of interest to the ACP community. However, I have major concerns regarding the quantification of measured species, as detailed in my comments below, which should be addressed before publication.

General comments

How much did the instrument sensitivity (measured as sensitivity to formic acid) change over time? Were any patterns observed in this change over time?

Very little deviation in the formic acid calibrations was observed. The mean average sensitivity was 30.66 $\pm$ 1.90 (1$\sigma$) Hz/ppt. We believe the sensitivity changes due to

reagent ion changes are minimal as mean average I⁻ + I.H₂O⁻ counts were high (3.52x10⁶ +/- 5.2x10⁵ (1σ) Hz).

The following text has been added:

"Very little deviation in the formic acid calibrations was observed. The mean average sensitivity was $30.66 \pm 1.90$ (1σ) Hz/ppt."

Specific comments:

Line 129: the authors describe how they minimized losses to the sample line. Did they then characterize the losses? What were they?

The inlet was very short (1m) with a fast flow rate (15 slm). Whilst loses for the exact species here were not characterised themselves, we have performed appropriate tests using nitric acid (whose partitioning to and from inlet walls the CIMS is particularly sensitive towards) with this type of inlet in the past (Bannan et al. 2014). We assume minimal sample line losses based on this previously observed behaviour.

Lines 159-161: the authors state that they "feel" this calibration method works well, but thats not very convincing, also considering that the results differ by 58% compared to the other calibration method applied. They authors then state that they consider this 58% their measurement uncertainty. A few points regarding this are that: 1) the measurement ncertainty is not mentioned anywhere else in the paper but should be quantitative results should be stated with a measurement uncertainty 2) other variables are expected to increase total uncertainty, including inlet and background effects and changes in instrument sensitivity. These other factors should be included in the total measurement uncertainty)

This comment has been addressed in the response to reviewer 1.

Line 164-166: do these previous studies ensure a 100% conversion efficiency (as currently stated) or do they assume it? If they ensure it, how so? If they assume it, what is the justification?

[Figure]

As discussed in the text previous studies assume 100% yield of ClNO$_2$ based on (Os-thoff et al. 2008; Kercher et al. 2009). This assumption is based on empirical data from a range of sources summarised in Finlayson-Pitts (2003).

Line 190 (loss processes of Cl): Why do the authors not include the VOC + Cl loss mechanism? (Presumably, this is how the observed ClOVOC are formed.)

Whilst the VOC + Cl reaction was mentioned in the text it was not included in the calculation has been updated to include the VOC + Cl reaction. In addition, the VOC loss term was not adequately described. These were mistakes and the following clarifications have been added to the text:

"The individual kCl+VOC are taken from the NIST chemical kinetics database.

$$[Cl]_{SS} = \frac{2J_{Cl_2}[Cl_2]+J_{ClNO_2}[ClNO_2]+J_{HOCl}[HOCl]+J_{ClOVOC}\sum[ClOVOCs]}{k_{O_3+Cl}[O_3]+k_{CH_4+Cl}[CH_4]+\sum_{n=0}^{i}[VOC]_i}$$

As methane was not measured, an average concentration was taken from ECMWF Copernicus atmosphere monitoring service (CAMS). VOC concentrations were approximated by applying representative VOC:benzene ratios for the UK urban environment (Derwent et al. 2000) and applying those to a typical urban UK benzene:CO ratio (Derwent et al. 1995) where CO was measured at the Whitworth observatory. The VOC:benzene ratios are scaled to the year of this study to best approximate ambient levels (Derwent et al. 2014). The calculated benzene:CO ratio is in good agreement with a Non-Automatic Hydrocarbon Network monitoring site (Manchester Piccadilly) approximately 1.5 km from the measurement location indicating that the approximation made here is reasonably accurate. The ratios assume traffic emissions are the dominant source of the VOCs as is assumed here."

Line 203-206: Have the authors checked whether photolysis of other ClOVOC are available in other models/databases, e.g. the JPL kinetics database

https://jpldataeval.jpl.nasa.gov/? How do the photolysis rates compare?

We have performed a literature survey and checked the JPL kinetics database but could not find a more suitable photolysis rates for ClOVOC than the one used. We have added the following clarification to the text

"As many of the identified species here do not have known photolysis rates, we approximate the photolysis of methyl hypochlorite $J_{CH3OCl}$ for all ClOVOCs as it is the only available photolysis rate for an oxygenated organic compound containing a chlorine atom provided by the TUV model and no other more suitable photolysis rate could be found elsewhere e.g. the JPL kinetics database."

Line 236-238: please justify not accounting for LOD when calculating statistics on the observed concentrations

We have used to simple average as levels of the observed species fall below their LODs.

[Figure]

Fig 1. Signal enhancement for $Cl_2$ due to $P_{H2O}$ in the IMR

**Fig. 1.**

[Figure]

Fig. 2. Summary of all $Cl_2$ instrument backgrounds. Left panel shows the magnitude of the background. Right panel shows the frequency distribution of background measurements. The outliers of the first plot are the inclusion of the e-folding time of the signal when the background begins. These points are not used for the calculation of the backgrounds.

Fig. 2.

[Figure]

Fig 3. I.H$_2$O vs RH. No correlation is observed

**Fig. 3.**

[Figure]

Fig 4. Cl$_2$ and C$_2$H$_3$O$_2$Cl (left). Global Radiation (right). I.H$_2$O (right). The water cluster signal is stable when the signals for Cl$_2$ and C$_2$H$_3$O$_2$Cl change.

**Fig. 4.**

---

## Author Response (AR2)

**Response to Reviewers**

Again we thank the reviewers for their further comments which are addressed below. The response to each point immediately follows each comment and is coloured red. Updated quotations from the manuscript are in blue. Quotes from the original text are in green.

**Reviewer 2.**

Review of "Observations of organic and inorganic chlorinated compounds and their contribution to chlorine radical concentrations in an urban environment in Northern Europe during the wintertime." by M. Priestley et al. ACP-2018-236

**Anonymous Referee Comments**

The improvements to the manuscript provide a sufficient change relative to previous versions to warrant publication of this manuscript to ACP. I have only minor comments which should be addressed prior to publications.

**Detailed Comments:**

I still remain apprehensive about the robustness of the zero determinations. The authors cite stability in instrumental mass calibration parameters and lack of IH2O changes as evidence of a stable background. The background however is a function of what has been previous sampled as well as instrumental parameters. As to figure 3 in the response, the authors should color the data by date/time and the correlations with ambient humidity will stand out. The jumps observed and scatter are most likely due to changes in tuning or sensitivity throughout the project, e.g. single ion current adjustments. I am happy to concede as the authors clearly state how backgrounds were performed in the manuscript for the readers.

As described in the response we recognise the limitations of the background method. We feel the 20 minute background time is a good compromise between reducing data capture losses and being sufficiently long enough to remove the effects of any species that are still present in the instrument.

I would like to see more information of the actual sensitivities included in the body of the manuscript, perhaps in the individual molecule sections in 3.1…

The calibration factors of the Cl species have been added to the text.

"The $Cl_2$ calibration factor is 4.6 Hz ppt$^{-1}$."

"As all chlorinated VOCs we observe are oxygenated we assume the same sensitivity found for 3-chloropropionic acid (10.32 Hz ppt$^{-1}$)"

"$ClNO_2$ was calibrated by the method described by Kercher et al. (2009) with $N_2O_5$ synthesised following the methodology described by Le Breton et al. (2014) giving a calibration factor of 4.6 Hz ppt$^{-1}$."

"We calibrate HOCl using the methodology described by Foster et al. (1999) giving a calibration

factor of 9.22 Hz ppt$^{-1}$."

If the detection limit for CH2Cl2 is 143 ppb and CHCl3 is 11ppb you could at most be missing 150ppb of Cl from those sources. How would that effect your budget?

We have not considered these species as sources of Cl as their photolysis rates are negligible at ground level as provided by the TUV model (described in section 2.2).

In the conclusion you should be explicit in saying those levels were below detection limit and removing the phrase "were not detected".

This distinction has been made more explicit by replacing the text in green with the text in blue

No aliphatic or polychlorinated species were detected, although the ToF-CIMS with I$^-$ is sensitive towards them e.g. methyl chloride ($CH_3Cl$) dimethyl chloride ($CH_2Cl_2$) and chloroform ($CHCl_3$).

Although the ToF-CIMS with I$^-$ is sensitive towards chlorinated and polychlorinated aliphatic compounds e.g. methyl chloride ($CH_3Cl$) dimethyl chloride ($CH_2Cl_2$) and chloroform ($CHCl_3$), their concentrations were below the detection limit.

On page 9, section 3.1 you are comparing CL2 observations to C2H4O5 and C10H14O4 to imply photochemical formation or lack thereof. How can you interpret the source of a compound that you have not identified and only have a molecular formula for?

OVOCs have a variety of primary and secondary sources, so differences in their diurnal profiles are expected. Whilst we do not know the identities unequivocally, the behaviour of $C_2H_4O_5$ is similar to $Cl_2$, $O_3$ production and the incidence of direct solar radiation, suggesting photochemistry whereas $C_{10}H_{14}O_4$ did not exhibit any of these behaviours. We make the assumption that a highly oxidised molecule such as $C_2H_4O_5$ with a high O:C ratio is more likely to be the product of photo-oxidative processes than the result of direct emission.

I do not see why the section on the newer ClNO2 calibration method is included here if the conclusion is that the approximations are incorrect and the sensitivity determined is wrong. Why not just drop that portion, use the Kercher 2009 method here and work on the method for a future publication?

We accept that the Kercher method is well established and reliable and so use it as our calibration method. We feel the TF-CIMS method is of interest to the community who, as well as ourselves, may also want to develop it further.

In your response, page 2. Where does the idea that formic acid has one of the strongest humidity dependences known for iodide adduct CIMS come from?

We make this observation based on our experience of calibrating many organic and inorganic compounds.

Abstract line 30, add space after contributing
Page 3, line 87, There seems to be a word missing in the last sentence.

Page 5 and 6, remove the quotes from this paragraph.

These typos have been corrected